# Unlocking ultra-high holographic information capacity through nonorthogonal polarization multiplexing

Jie Wang[1,2,6], Jin Chen [1,6], Feilong Yu[1,6], Rongsheng Chen[1], Jiuxu Wang[1], Zengyue Zhao[1], Xuenan Li[1], Huaizhong Xing[2], Guanhai Li [1,3,4,5] ✉, Xiaoshuang Chen[1,3,4,5] & Wei Lu [1,3,4,5]

Contemporary studies in polarization multiplexing are hindered by the intrinsic orthogonality constraints of polarization states, which restrict the scope of multiplexing channels and their practical applications. This research transcends these barriers by introducing an innovative nonorthogonal polarization-basis multiplexing approach. Utilizing spatially varied eigen-polarization states within metaatoms, we successfully reconstruct globally nonorthogonal channels that exhibit minimal crosstalk. This method not only facilitates the generation of free-vector holograms, achieving complete degrees-of-freedom in three nonorthogonal channels with ultra-low energy leakage, but it also significantly enhances the dimensions of the Jones matrix, expanding it to a groundbreaking $10 \times 10$ scale. The fusion of a controllable eigen-polarization engineering mechanism with a vectorial diffraction neural network culminates in the experimental creation of 55 intricate holographic patterns across these expanded channels. This advancement represents a profound shift in the field of polarization multiplexing, unlocking opportunities in advanced holography and quantum encryption, among other applications.

Polarization orthogonality, a fundamental principle in photonics, demands that the inner product of two output fields, $\vec{u}_1(x,y)$ and $\vec{u}_2(x,y)$, equals zero. This orthogonality, stemming from polarization bases $\hat{p}_1$ and $\hat{p}_2$, confines polarization multiplexing to a mere two channels, dictated by a $2 \times 2$ Jones matrix[1,2]. While this orthogonality ensures perfect isolation in applications such as polarization imaging and information encryption[3–10], it inherently limits the maximum number of multiplexing channels free from cross-interference. Existing methods, like space- or time-division techniques[11–17], compromise on spatial or temporal resolution, and typically can only expand to four orthogonal polarization channels. For instance, conventional

configurations using bulk optical elements like polarizers and waveplates represent this limitation[18]. Attempts to incorporate additional dimensions[19–23], such as wavelength or orbital angular momentum, typically exacerbate issues of crosstalk and noise. Even with advanced eigen-polarization modulation techniques, the reliance on orthogonal polarization bases persists, constraining applications like dynamic holography and information transmission within a narrow scope[17,24–26].

Recent advancements in metasurface technology have significantly expanded the potential for holographic information capacity. For instance, Bao et al. cascaded two single-layer metasurfaces to construct a spatially varying Jones matrix, enabling the manipulation

[1]State Key Laboratory of Infrared Physics, Shanghai Institute of Technical Physics, Chinese Academy of Sciences, 500 Yu-Tian Road, Shanghai 200083, China. [2]College of Physics, DongHua University, 2999 North Renmin Road, Shanghai 201620, China. [3]Hangzhou Institute for Advanced Study, University of Chinese Academy of Sciences, No.1 SubLane Xiangshan, Hangzhou 310024, China. [4]University of Chinese Academy of Science, No. 19 Yuquan Road, 100049 Beijing, China. [5]Shanghai Research Center for Quantum Sciences, 99 Xiupu Road, Shanghai 201315, China. [6]These authors contributed equally: Jie Wang, Jin Chen, Feilong Yu. ✉e-mail: ghli0120@mail.sitp.ac.cn

of the amplitude and phase of light for intricate holographic applications[8]. Furthermore, Wang et al. demonstrated high-efficiency metasurfaces capable of complex vectorial holography using polarization multiplexing, showing that engineered metasurfaces can achieve high-density information encoding[9]. Additionally, Xiong et al. introduced the concept of engineered noise to break the fundamental limit of polarization multiplexing capacity, demonstrating up to 11 independent holographic images using a single metasurface[17]. These studies demonstrate the possibility of breaking traditional limitations by employing innovative strategies, such as engineered noise and complex light manipulation techniques, to enhance polarization multiplexing. Despite these advancements, the number of achievable independent channels remains a critical area for further research, as the lack of full exploitation of metasurfaces restricts high-capacity holographic applications, particularly in fields requiring extensive data encoding and secure information transmission.

Here, our work pioneers a nonorthogonal polarization-basis multiplexing technique by engineering spatially variable eigen-polarization states $\hat{p}_1(x,y)$ and $\hat{p}_2(x,y)$ at a subwavelength scale using metaatoms. We meticulously control the local eigen-polarization of each metaatom, thereby achieving a unique collective effect across multiple nonorthogonal polarization channels. This method, deviating from the norm, permits a nonzero product of $\vec{u}_1^*(x,y)$ and $\vec{u}_2(x,y)$ at each metaatom, culminating in a precise overall polarization output. Our approach's efficacy is demonstrated both theoretically and experimentally by showcasing three holograms cycled through asymmetric polarization channels, achieving full degrees-of-freedom coverage without compromising spatial, temporal, or other dimensions. Furthermore, we introduce a controllable local eigen-polarization modulation mechanism that expands the Jones matrix dimensionality to 10×10. By employing a vectorial diffraction neural network (VDNN)[27], we optimize multiplexing efficiency and reduce crosstalk, as evidenced by a set of 55 holograms integrating 10 input and 10 output polarization states. Notably, there is no compromise of spatial, temporal, or other dimensions in the realization of three-channel nonorthogonal polarization multiplexing. For cases involving more than three channels, the spatial dimension along the propagation direction is adopted in this work. Our work represents a substantial advance in polarization multiplexing channels and it also offers a robust and scalable solution for ultra-high capacity holographic multiplexing. This nonorthogonal strategy promises new possibilities for dynamic holography and advanced information transmission in the realm of photonics.

## Results
### Design principle
The polarization manipulation capabilities of a single-layer metaatom are elegantly encapsulated by the Jones matrix, represented as:

$$J = \begin{bmatrix} A & B \\ C & D \end{bmatrix} = [\hat{e}_1, \hat{e}_2]^{-1} \begin{bmatrix} \Lambda_1 & 0 \\ 0 & \Lambda_2 \end{bmatrix} [\hat{e}_1, \hat{e}_2] \tag{1}$$

Here, A, B, C, and D govern the complex amplitude control of both eigen and cross channels in standard polarization representation. The eigen-polarizations $\hat{e}_1$ and $\hat{e}_2$, along with the eigenvalues $\Lambda_1$ and $\Lambda_2$, define the polarization control. Equation (1) highlights that the polarization control is simultaneously constrained by space-time reciprocity and the inherent form of the eigen-polarization.

Traditionally, using orthogonal basis vectors $|H\rangle$ and $|V\rangle$, the Jones matrix is decomposed into components such as $|H\rangle\langle H|$, $|H\rangle\langle V|$, etc. Due to the reciprocity of single-layer metasurfaces, it's established that B = C, thus limiting the polarization control to three degrees of freedom.

However, this approach simplifies the complex relationship between the local eigen-polarization control $J$ of a metaatom and the global output polarization from an input. For in-plane symmetric metaatoms, the inherent linearity of the eigen-polarization constrains input-output combinations to linear orthogonal polarization bases in conventional scenarios.

Our work transcends these limitations by modulating the local linear eigen- polarizations of metaatoms, thus constructing globally nonorthogonal output polarization states (see Fig. 1a). This fine-tuning of each metaatom's local eigen-polarization results in distinct collective effects on multiple nonorthogonal output polarization channels.

For arbitrary input $\boldsymbol{p}_i$ and output $\boldsymbol{p}_j$($i, j$=1, 2, ..., n), the responses $O_{ij}(x,y) = \langle \boldsymbol{p}_j|J|\boldsymbol{p}_i \rangle$ of metaatoms at different positions (x, y) differ from each other. Combining with Eq. (1), we have $O_{ij}(x,y) = \boldsymbol{p}_j^\dagger \cdot [\hat{e}_1, \hat{e}_2]^{-1} \cdot \Lambda \cdot [\hat{e}_1, \hat{e}_2] \cdot \boldsymbol{p}_i$, where † represents the transpose and conjugate operation. Since the system is unitary, $[\hat{e}_1, \hat{e}_2]^{-1} = [\hat{e}_1, \hat{e}_2]^\dagger$, thus:

$$\begin{aligned} O_{ij}(x,y) &= ([\hat{e}_1, \hat{e}_2] \cdot \boldsymbol{p}_j)^\dagger \cdot \Lambda \cdot ([\hat{e}_1, \hat{e}_2] \cdot \boldsymbol{p}_i) \\ &= \boldsymbol{p}_j'^\dagger \cdot \Lambda \cdot \boldsymbol{p}_i' \end{aligned} \tag{2}$$

where $\boldsymbol{p}_i'$, and $\boldsymbol{p}_j'$ include the role of eigenvectors in their interaction with $\boldsymbol{p}_i$ and $\boldsymbol{p}_j$.

At this point, we have derived the fundamental equation $O_{ij}(x,y)$ for the modulation of incident/output polarization along with the metaatoms, which serves as the basis for extending arbitrary polarization channel modulation. By incorporating this equation into the tools of machine learning or neural networks, it offers an efficient way to address the challenge of maximizing the isolation between channels that jointly modulate global perturbations and local controls under limited degrees of freedom. In this situation, we can achieve desired nonorthogonal multiplexing channels for any combined input and output polarization pairs.

To demonstrate this, we choose circular polarization basis vectors to rewrite Eq. (2) and get Eq. (3):

$$\begin{aligned} O_{ij}(x,y) &= \langle \boldsymbol{p}_j|J_{circ}|\boldsymbol{p}_i \rangle \\ &= (a_{3j}^*\langle L| + a_{4j}^*\langle R|) \cdot (J_{eigen} + J_{con}) \cdot (a_{1i}|L\rangle + a_{2i}|R\rangle) \\ &= a_{3j}^* \cdot a_{1i} \cdot A_{LL} + a_{3j}^* a_{2i} \cdot B_{LR} + a_{4j}^* a_{1i} \cdot C_{RL} + a_{4j}^* a_{2i} \cdot A_{RR} \end{aligned} \tag{3}$$

In this scenario, we can decompose the metaatom's matrix into eigen and conjugate components $J_{eigen} = \begin{bmatrix} A_{LL} & 0 \\ 0 & A_{RR} \end{bmatrix}$ and $J_{con} = \begin{bmatrix} 0 & B_{LR} \\ C_{RL} & 0 \end{bmatrix}$, as shown in Fig. 1b, where each element represents the circular-based modulation channel, $A_{LL}/A_{RR}$ and $B_{LR}/C_{RL}$ are co- and cross-polarization channel respectively. $|\boldsymbol{p}_i\rangle = a_{1i}|L\rangle + a_{2i}|R\rangle$ and $\langle \boldsymbol{p}_j| = a_{3j}^*\langle L| + a_{4j}^*\langle R|$, where $a$ is decomposition factor, * denotes conjugation, and $|L\rangle/|R\rangle$ are left/right circular polarization. This decomposition enables us to explore various combinations of incident and output polarizations, showing how the global response of the metasurface and the local response of the metaatoms are intricately cross-coupled.

Intuitively, the metaatom's response based on eigen-polarization modulation can be denoted by a matrix $O(x, y)$ (Eq. (4)), in which different combinations of input and output polarizations are selective extractions of this response matrix. The out-of-plane symmetry leads to physical transport reciprocity in our system, which gives the response matrix the form of a symmetric matrix, i.e., the upper (lower) triangular form, the relevant discussion can be seen in

Supplementary Note 1.

$$O(x,y) = [\boldsymbol{p_1}, \boldsymbol{p_2}, \ldots, \boldsymbol{p_n}]^\dagger \cdot J(x,y) \cdot [\boldsymbol{p_1}, \boldsymbol{p_2}, \ldots, \boldsymbol{p_n}]$$

$$= \begin{bmatrix} O_{11}(x,y) & O_{12}(x,y) & O_{13}(x,y) & \ldots & O_{1n}(x,y) \\ \sim & O_{22}(x,y) & O_{23}(x,y) & \ldots & O_{2n}(x,y) \\ \sim & \sim & O_{33}(x,y) & \ldots & O_{3n}(x,y) \\ \vdots & \vdots & \vdots & \ddots & \vdots \\ \sim & \sim & \sim & \ldots & O_{nn}(x,y) \end{bmatrix} \quad (4)$$

The global response of metasurface for nonorthogonal polarization multiplexing is then the integral of Eq. (4) with respect to position: $\Omega = \iint O(x,y)dxdy$. In this way, the change of the response matrix due to the tuning of the input-output polarization will be fully attributed to the modulation effect of the eigenvectors on the input-output polarization. According to this integral formula and the diffraction process, we can build an efficient network and optimize the parameters so that the response matrix $\Omega$ can optimally extract different channels selectively. Details can be found in Supplementary Note 2.

It's worth noting that we utilize the spatially varied eigen-polarization states of the metaatoms to successfully reconstruct globally nonorthogonal channels that exhibit minimal crosstalk. This leads to a modulation capability that is no longer limited to symmetric channels ($\Omega_{ij}, i = j$), but can also be extended to asymmetric channels ($\Omega_{ij}, i \neq j$). Equation (4) represents a pivotal development in our study. It enables the reconstruction of the control matrix over various non-orthogonal input-output polarization channels, significantly expanding the conventional dimension of the Jones matrix.

## Experiments and characterization of the metasurface

Employing the methodology delineated, we executed a series of sample designs to empirically validate our proposed nonorthogonal polarization-basis multiplexing approach. This included an array of tri-fold cyclic nonorthogonal linear polarization multiplexes, tri-fold cyclic nonorthogonal circular and elliptical polarization multiplexes, and an extensive suite of 55 diverse nonorthogonal polarization multiplexes.

In the implementation of our triple nonorthogonal linear polarization asymmetric holography model, we simplified the metasurface

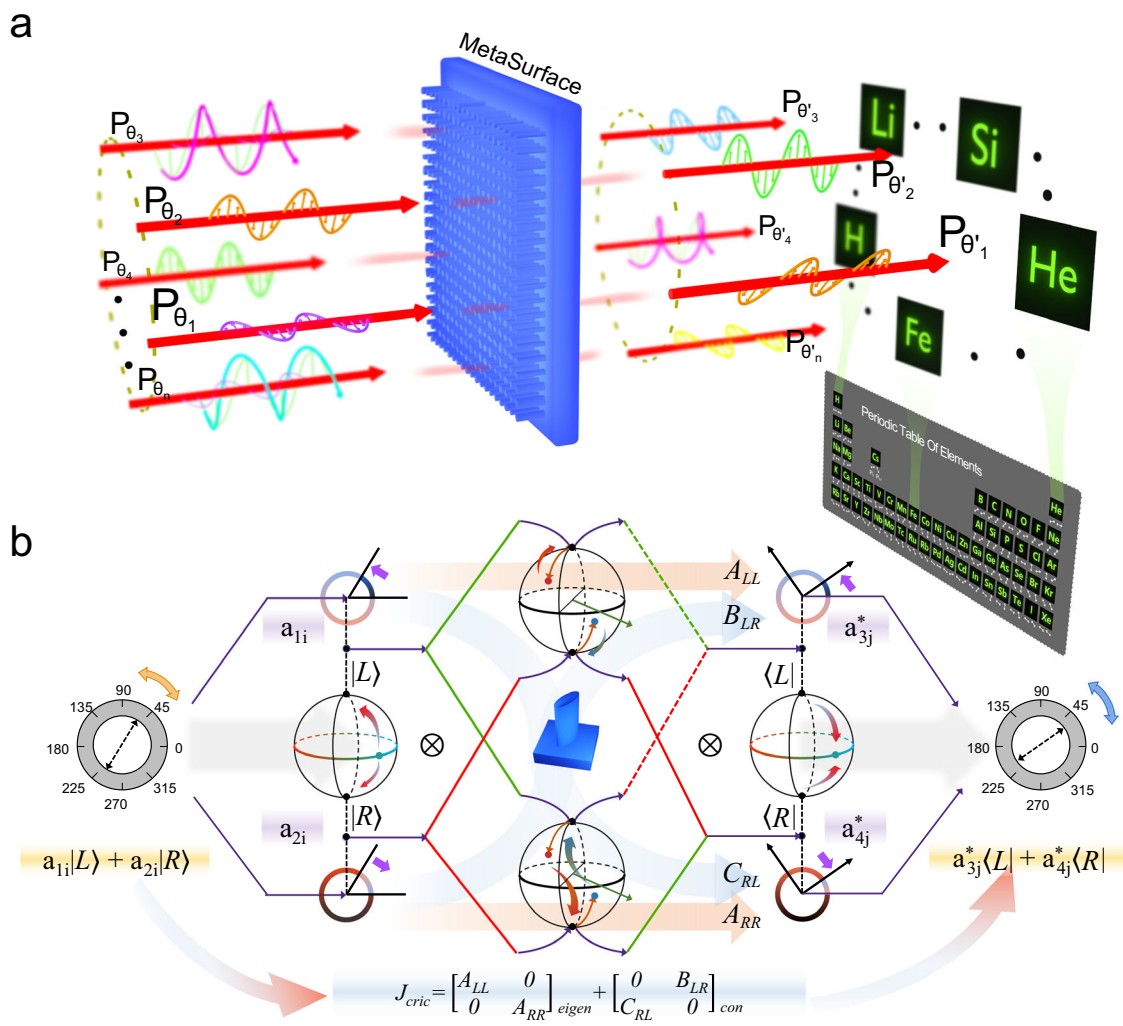

**Fig. 1 | Conceptualization and theoretical framework of the dynamic holographic metadevice. a** Illustration of a dynamically polarized holographic metadevice. It is engineered to generate holograms for each of the 55 arbitrary polarization channels, analogous to the elements in the periodic table. **b** Schematic of the controllable eigen-polarization modulation mechanism. An arbitrary input polarization is initially decomposed into circular polarization bases |L⟩ and |R⟩. For demonstrative purposes, linear polarization is selected. Upon interaction with a local metaatom, the polarization undergoes transformations in both eigen and conjugate channels, namely $A_{LL}$, $B_{LR}$, $C_{RL}$, and $A_{RR}$. Following this, a dynamically rotating polarizer is applied, assigning $a_{ij}$ manipulation to the ⟨L| and ⟨R| channels, respectively. This process enables the modulation of local eigen-polarization at the metaatom level, facilitating global polarization multiplexing. The cumulative and coherent interference of various metaatoms at the imaging plane realizes the envisioned nonorthogonal polarization multiplexing.

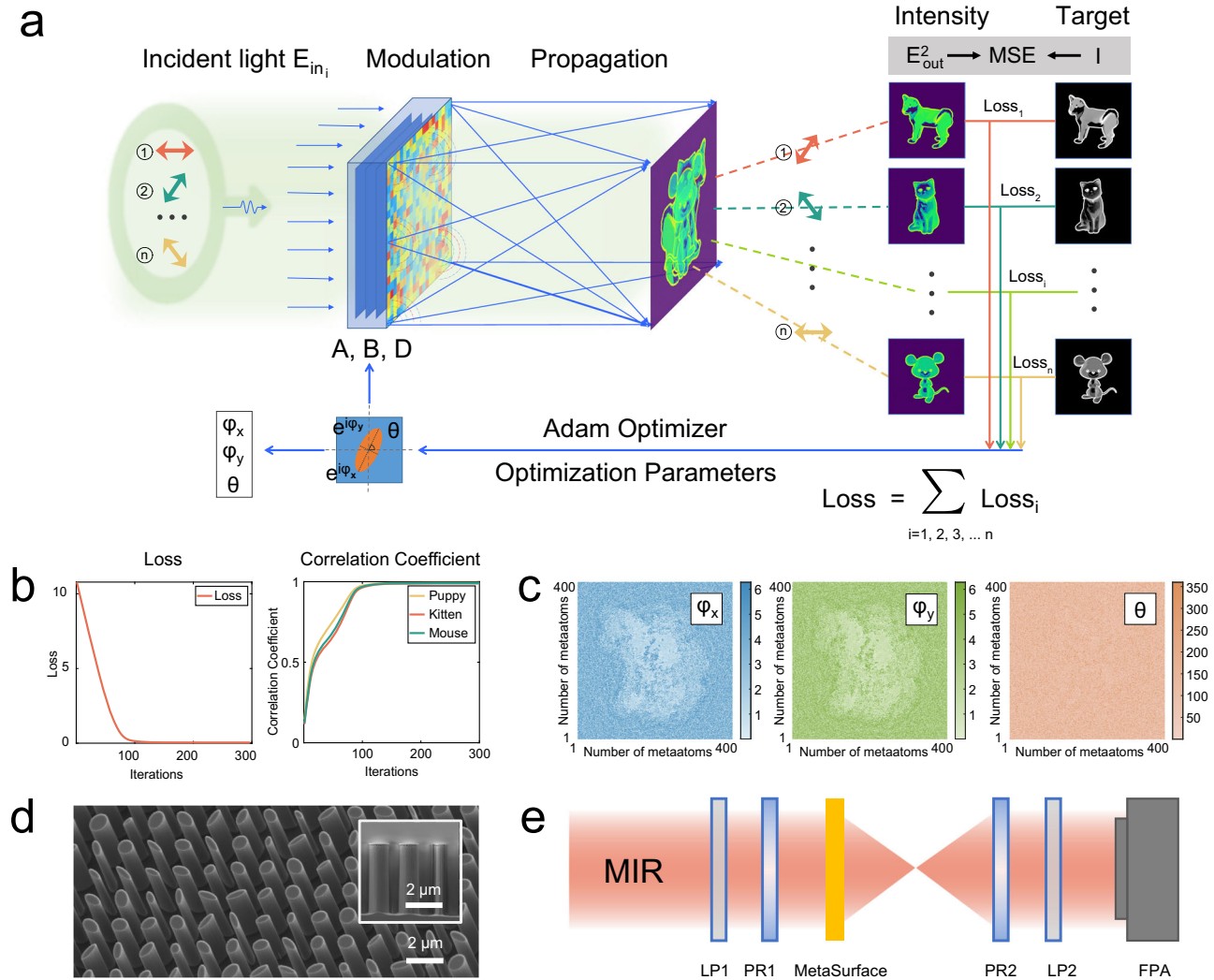

**Fig. 2 | Vectorial diffraction neural network design, sample fabrication, and experimental setup. a** Conceptual model of vectorial diffraction neural network illustrating the training process. Here, different polarized lights constitute the input layer, the metasurfaces function as the hidden layers, and the image plane serves as the output layer. **b** Training outcomes for the nonorthogonal linear polarization channel. This section highlights the loss maps and the correlation coefficients that measure the congruence between the output and target patterns. **c** Anisotropic response and orientation of the metaatoms within the systematically arranged metadevice. **d** Scanning electron microscope images, offering both oblique and cross-sectional views of the metadevice. Each image includes a scale bar of 2 μm to provide a sense of scale. **e** Depicts the optical characterization system used for evaluating the performance of the designed metadevices. LP stands for linear polarization, PR for phase retarder, and FPA for focal plane array.

to exhibit a transmittance of 1. The corresponding diffraction neural network was configured with a single hidden layer for streamlined efficiency. It should be noted that the definition of the diffraction neural network[28–30] follows that outlined in ref. 27. However, our approach differs in that we integrate the polarization dimension of light into a single-layer metasurface, rather than using a series of cascading optical elements. The optimization is conducted using electronic neural networks[31]. As depicted in Fig. 2a, the training process is illustrated, highlighting the incident polarization, output polarization, and the targeted holographic pattern. Operating at a wavelength of 3μm, we precisely calibrated incident polarization angles at 0, 60, and 120 degrees to correspond with output polarization angles of 60, 120, and 0 degrees, respectively. This calibration resulted in the generation of gray-scale holographic images: a puppy, a kitten, and a mouse. The distance between the metasurface layer and the output layer was meticulously maintained at 500 μm. The loss for each target pattern is evenly distributed with its respective weight during the optimization process, promoting a fair allocation of energy across all patterns. More details can be found in Supplementary Note 2.

Figure 2b offers insight into the neural network training, showcasing the loss profile and the variation in correlation coefficients across channels as a function of iteration numbers. Over successive iterations, the three patterns gradually converge towards the target patterns at varying rates, ultimately attaining a nearly flawless alignment as the iteration count rises. The vectorial diffraction neural network's output results, including the parameters $\varphi_x$, $\varphi_y$, and the metasurface orientation $\theta$, are elucidated in Fig. 2c. The metasurface consisted of a 400 × 400-pixel array, each with a period of 1.5 μm. The nano-elliptic metaatoms, made of pure silicon, have a height of 4 μm, the length and width of the metaatom varied between 0.3 and 1.2 μm. More details are in Supplementary Note 3. In Fig. 2d, scanning electron microscopy images provide both oblique and cross-sectional views of the metasurface, showcasing the robustness of the fabrication process and underscoring the precision achieved in metasurface manufacturing. The scale bar is 2 μm. Lastly, Fig. 2e delineates the experimental setup, where the camera captures the holographic pattern following the transmission of infrared light through the polarizer, metasurface, and analyzer. The phase retarder was omitted in the linear polarization

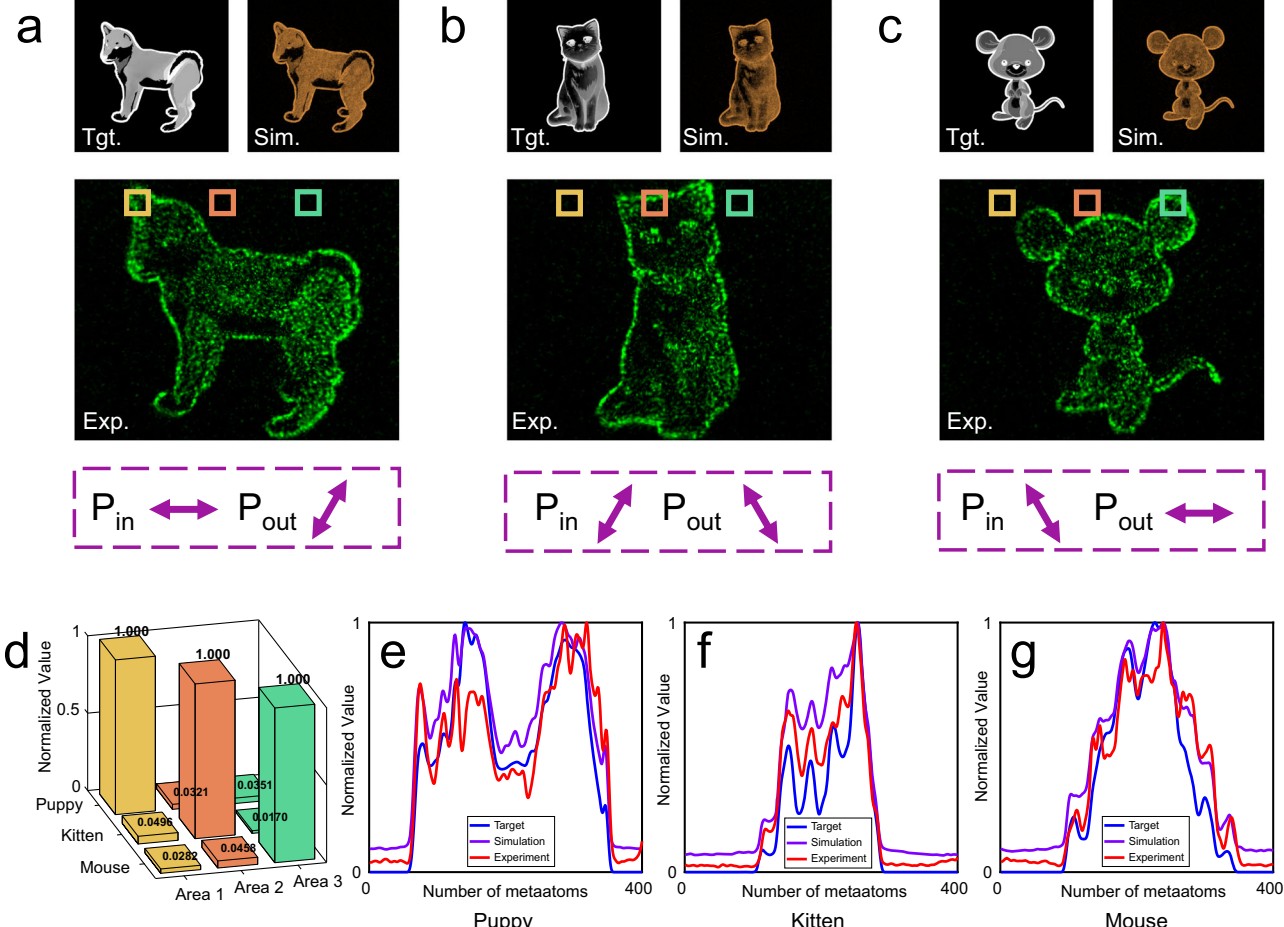

**Fig. 3 | Experimental results and comprehensive data analysis. a–c** Training targets (upper left), simulation results (upper right), and experimental results (middle) for the tri-fold cyclic nonorthogonal polarization holography. The bottom section of each panel, highlighted in purple boxes, denotes the distinct polarization channels. The images illustrate the holographic patterns: a puppy in (**a**) with polarization transitioning from 0 to 60 degrees, a kitten in (**b**) with polarization from 60 to 120 degrees, and a mouse in (**c**) with polarization from 120 to 0 degrees, all within the nonorthogonal polarization channel. **d** Crosstalk analysis. This panel presents a detailed crosstalk analysis, using selected metaatoms from experimental results (**a**)–(**c**), as indicated by yellow (Area 1), orange (Area 2), and green (Area 3) boxes. This analysis quantifies the interference between different polarization channels. **e–g** Longitudinal intensity statistics derived from the target patterns, simulation results, and experimental data, corresponding to (**a**), (**b**), and (**c**), respectively. These statistics provide a quantitative comparison between the intended patterns and both the simulated and actual experimental outcomes.

experiment. Further details on simulations and experiments can be found in the Methods section.

Figure 3a–c displays the outcomes of the tri-fold cyclic non-orthogonal linear polarization asymmetric holography experiments. These figures are systematically organized, featuring the training targets (top left), simulation results (top right), and experimental data (middle). The asymmetric polarization channels are distinctly highlighted in purple boxes at the bottom of each figure. Notably, these results showcase vivid and crosstalk-free asymmetric holograms achieved through nonorthogonal polarization multiplexing. This alignment serves as a testament to the robustness and efficacy of the proposed methodology. Supplementary Note 4 illustrates the degrees of freedom in the conventional Jones matrix for metasurface design.

To further quantify crosstalk among these hologram channels, we adopted partitioned summation statistics across three randomly chosen, disjoint regions, as detailed in the Supplementary Note 5. Figure 3d presents the crosstalk computations, indicating an almost negligible interference between hologram channels, thus affirming the method's precision.

Further statistical analysis is provided in Fig. 3e–g, where we juxtapose the experimental results with both target and simulated images. The employed statistical approach involves summing and normalizing each image along its longitudinal axis. It can be seen that the measured results agree well with the simulations. Moreover, the curve alterations exhibit a high degree of consistency with the target variations, emphasizing the fidelity and accuracy of our experimental findings. This quantitative evaluation not only substantiates the alignment of our results with the intended outcomes but also underscores the reliability of our experimental approach. The holographic efficiency is defined as $\eta = \frac{E_{holo}}{E_{total}}$, where $E_{holo}$ refers to the measured total energy of the hologram patterns, and $E_{total}$ refers to the total incident energy received by the metasurface area. The measured holographic efficiencies are 26.04% ($0° \to 60°$ polarized channel), 25.25% ($60° \to 120°$ polarized channel), and 27.83% ($120° \to 180°$ polarized channel) respectively.

To demonstrate the extensive engineering capabilities in creating asymmetric holograms across various nonorthogonal polarization channels, we developed tri-fold cyclic nonorthogonal holograms specifically designed for arbitrary circular and elliptical polarization channels. This process necessitated meticulous amplitude manipulation during the polarization layer mapping phase, as detailed in Fig. 4a.

For empirical validation, two distinct metasurface samples were rigorously designed and fabricated. The first sample was engineered to respond to incident light composed of x-, y-, and right-hand circularly

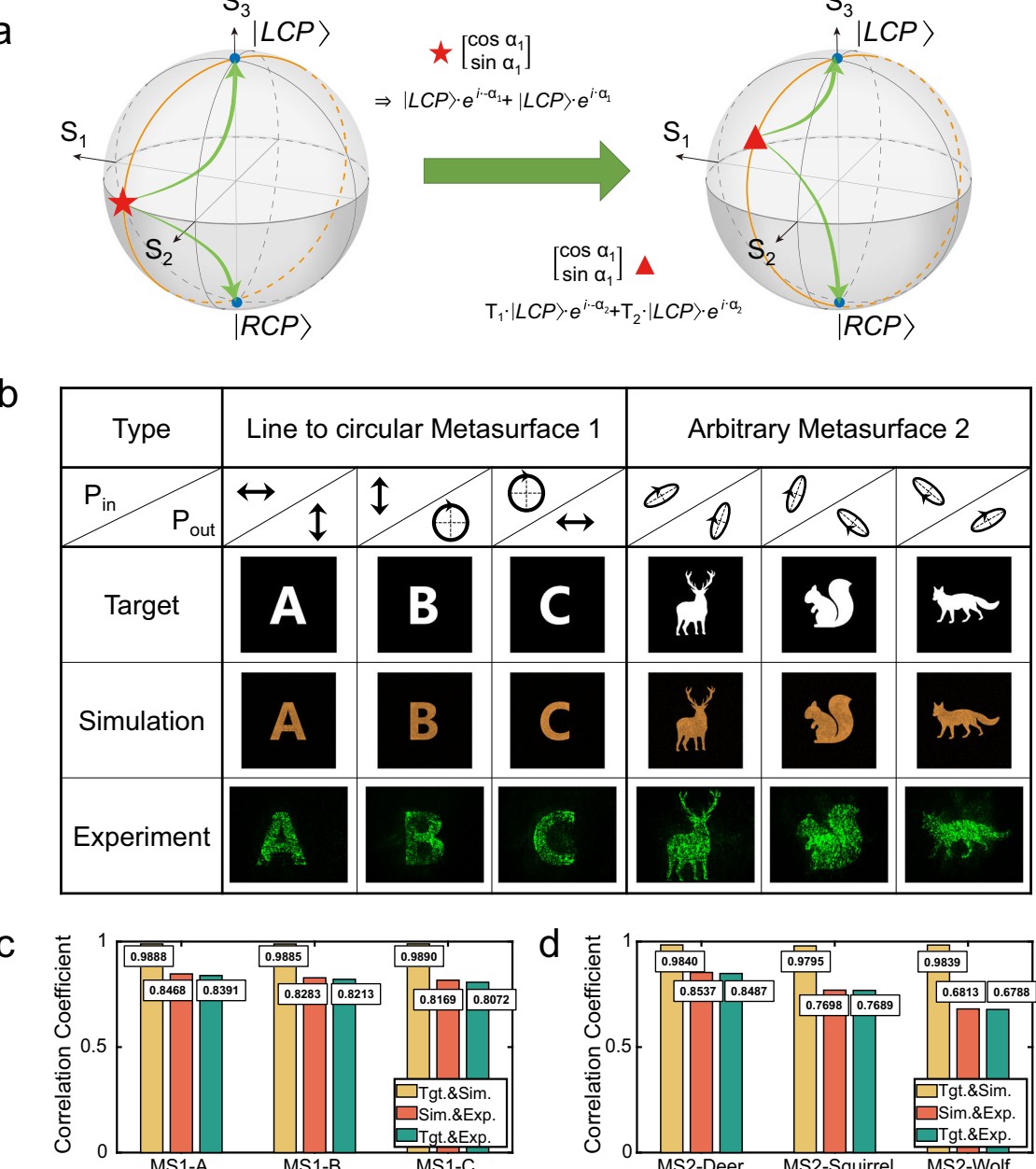

**Fig. 4 | Advanced nonorthogonal polarization multiplexing across diverse channels with optimal isolation. a** The operation mechanism that enables the extension of nonorthogonal polarization to a variety of forms. This is achieved through the strategic recombination and distribution of circular and elliptical polarization components, which facilitates the creation of a spectrum of nonorthogonal polarization channels. **b** Simulation and experimental results for two metasurfaces. It exemplifies the linear-circular polarization channel (using

Metasurface 1, MS1) and demonstrate the mutual conversion capabilities for arbitrary elliptical polarization (using Metasurface 2, MS2). The deer image has been designed using images from Flaticon.com. **c, d** Quantitative analysis of the correlation coefficients between target patterns, simulation outcomes, and experimental images for both circular and elliptical polarization channels. It highlights the exceptional isolation maintained between the various distinct polarization channels.

polarized components. This configuration was achieved by setting $\beta$ at $\pi/2$, and $\alpha$ at 0, 90, and 135 degrees within the Jones vector $\begin{bmatrix} \cos \alpha \\ \sin \alpha \cdot e^{i\beta} \end{bmatrix}$, resulting in holographic representations of the letters 'A', 'B', and 'C'. The second sample was tailored to process three nonorthogonal elliptical polarizations, with $\beta$ consistently at $\pi/3$, and $\alpha$ at 30, 70, and 140 degrees, producing holographic depictions of a deer, squirrel, and wolf, respectively. The training, simulation, and experimental results of these implementations are presented in Fig. 4b. Furthermore, the multiplexing of arbitrarily nonorthogonal linear polarizations is demonstrated in Supplementary Note 6.

To quantitatively assess the quality of the reconstructed images, we employed correlation coefficients between the reconstructed, target, and simulated images, as depicted in Fig. 4c, d. The correlation coefficient is defined as $corr(T,R) = \frac{\mathrm{COV}(T,R)}{\sqrt{D(T) \cdot D(R)}}$, where T and R denote the two images, COV (T, R) is their covariance, and D(T) and D(R) are their variances. Both metasurfaces were operated at a wavelength of 3 µm, with a pixel configuration of $400 \times 400$ and a diffraction distance of 500 µm. Despite the inherent challenges in metaatom determination and sample fabrication, the experimental results demonstrated a close alignment with the design targets. This congruence attests to the

feasibility and effectiveness of realizing arbitrary nonorthogonal polarization multiplexed holography. A nine-channel multiplexing with the introduction of wavelength is illustrated in Supplementary Note 7. The measured efficiencies of the Metasurface 1 are 19.13%, 22.16%, and 20.12%, corresponding to the letters 'A', 'B', and 'C', respectively, and the Metasurface 2 are 22.08%, 24.61%, and 21.4% in that order.

Building on the innovative nonorthogonal polarization multiplexing framework, we have developed a continuous polarization-based tunable Vectorial Diffraction Neural Network (VDNN). This groundbreaking approach is designed for the full exploitation of global polarization degrees of freedom, thereby significantly enhancing the information channel capacity. By integrating the controllable eigen-polarization modulation mechanism and introducing a subtle focal plane mismatch, we have successfully generated 55 distinct holographic patterns, as showcased in Fig. 5a.

In this system, ten linear polarizations (at angles 0°, 18°, 36°, 54°, 72°, 90°, 108°, 126°, 144°, 162°) are chosen for each input and output, cumulatively resulting in 55 multiplexing channels. These channels are systematically arranged in a lower triangular matrix of dimension 10 × 10, akin to a periodic table of elements, with each channel representing a unique matrix element, as depicted in Fig. 5a. Relevant design process is seen in Supplementary Note 8. The correlation coefficient matrix, presented in Fig. 5b, confirms the high degree of effective isolation achieved between these distinct channels.

Adhering to this methodology, we fabricated a metasurface comprising an array of 640,000 (800 × 800) metaatoms. The target pattern was tailored for experimental validation, with comprehensive details provided in the Supplementary Note 9. Experimental results are illustrated in Fig. 5c, where the input polarization is displayed at the bottom, the output polarization on the left, and the intersecting pattern depicting the experimentally measured channel results. The measured holographic efficiencies can be found in Supplementary Note 10. It's worth noting that the images in Fig. 5c are captured at slightly different spatial positions along the z-axis to ensure the best possible resolution and image quality for each holographic pattern. In addition, the non-uniform brightness distribution observed in some holographic patterns (e.g., Ni, Te, Fe) in Fig. 5c become more pronounced with an increase in the number of operating channels. This phenomenon occurs because, as the number of multiplexing channels increases, the repeated use of each metaatom also increases. Consequently, the control capacity of each metaatom may not fully meet the requirements, leading to changes in the brightness of the holographic patterns due to interference at the focal plane. To mitigate this effect, several strategies can be employed: 1. Increase the height of metaatoms: This can enhance the phase control range of each metaatom, thereby reducing brightness variations. 2. Use of combined metaatoms: Implementing complicated metaatoms with more design degrees of freedom can provide better control over the holographic output. 3. Adopt multiple layered

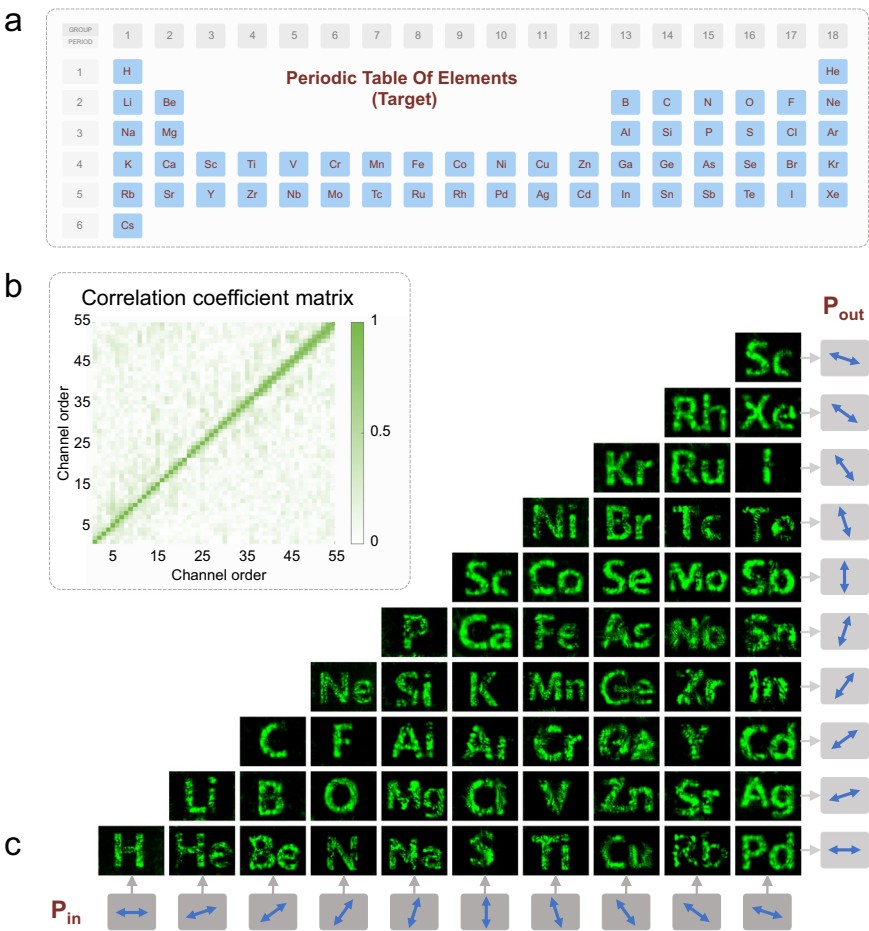

**Fig. 5 | Achievement of 55 holographic channels representing the periodic table of elements. a** Schematic illustration of 55 distinct elements selected from the periodic table, each serving as a target pattern for holograms across various orthogonal polarization channels. **b** Correlation coefficient matrix derived from the simulation results. This matrix exemplifies the exceptional global performance achieved in optimizing metaatoms for nonorthogonal polarization multiplexing, highlighting the accuracy and fidelity of our simulation process. **c** Experimental results of 55 nonorthogonal polarization channels. The arrows within the outer gray box signify the input and output polarizations. This representation emphasizes our method's ability to transcend conventional boundaries in polarization multiplexing, thereby demonstrating the superior capability of our approach in facilitating nonorthogonal polarization multiplexing across multiple channels.

metasurfaces: Utilizing multiple layers can distribute the control load more evenly across different metaatoms, improving uniformity. 4. Increase the number of metaatoms: Adding more metaatoms can enhance the overall control precision and reduce nonuniform brightness. These methods can improve the control degree of each metaatom more effectively and regulate the distribution of the holographic target pattern, thereby minimizing the nonuniformity in brightness.

## Discussion

The nonzero analytical solution of the derived formulas in Supplementary Notes 1 and 4 indicates that the largest rank of the coefficient matrix of the Jones matrix is 3. This means that for given metaatoms with similar profiles, the maximum number of channels for nonorthogonal polarization multiplexing without compromising other dimensions is three. Beyond three channels, crosstalk becomes significant, and different methods must be employed to extend the multiplexing channels. Our approach involves the use of a controllable local eigen-polarization modulation mechanism, as well as optimization through a vectorial diffraction neural network. Additionally, methods such as noise assistance, as described in ref. 17, can also be utilized to further extend the number of multiplexing channels. To achieve more multiplexing channels, it imposes a higher requirement on the metaatoms to possess stronger control abilities, such as enhanced phase coverage, more design degrees of freedom, etc. This necessitates metaatoms with larger depth-to-width ratios, higher refractive index differences, more complex shapes, or additional metasurface layers to expand their controlling capabilities. These factors can influence the efficiency, crosstalk, and the number of achievable multiplexing channels.

As for the potential to implement other types of functionalities, such as a continuous zoom lens by varying the input-output polarization, our method theoretically allows for such implementations. By precisely controlling the eigen-polarizations of the metaatoms and optimizing the metasurface's global response, we could potentially develop functionalities beyond holography, including adaptive lenses and other dynamic optical devices. This methodology offers the potential to further expand the Jones matrix to even higher dimensions. Such expansion facilitates the manipulation of more complex polarization states and channels. The compact and versatile nature of these metadevices, underpinned by the controllable eigen-polarization engineering mechanism, solidifies their role as a robust platform for advanced hyper-polarization dynamic holography.

In summary, this study marks a significant breakthrough in the field of photonics by pioneering a coupled recombination design that leverages local eigen-polarization degrees of freedom. The integration of this mechanism with a sophisticated vectorial diffraction neural network design has been instrumental in achieving a high level of polarization multiplexing holography, encompassing 55 distinct channels. This feat not only facilitates the creation of arbitrary polarization forms and asymmetric channels but also greatly expands the dimensional capacity of the Jones matrix through our controllable eigen-polarization modulation mechanism.

Furthermore, by harmonizing various multiplexing techniques, our strategy significantly boosts the capacity and security of information transmission while maintaining minimal crosstalk. The introduction of nonorthogonal polarization-basis multiplexing represents a pivotal development, establishing a robust foundation for future innovations in optical communications and quantum information sciences. This work not only exemplifies state-of-the-art scientific exploration but also opens new horizons for more secure and versatile channels in the ever-evolving domain of photonics.

## Methods
### Sample fabrication
The fabrication of the all-silicon metasurface begins with the deposition of a 50 nm chromium (Cr) layer onto a double-sided polished silicon wafer, utilizing electron beam evaporation. A photoresist layer is then spin-coated onto this chromium-coated wafer, followed by a baking process on a hot plate. The metasurface pattern is intricately defined using electron beam lithography (JBX-6300FS). Post-lithography, the samples are developed in a 300-MIF solution, rinsed thoroughly with deionized water, and dried. The fabrication is completed through the use of inductively coupled plasma (ICP) dry etching, targeting both the chromium and silicon layers. This etching process involves selective material removal using various gases, ensuring the precise realization of the desired metasurface configuration.

### Measurement procedure
The experimental setup involves the generation of mid-infrared light using the Electro MIR 4.8 laser system (Leukos laser). The light is first filtered to a wavelength of 3 μm with a 200 nm bandwidth, then passes through a polarizer and a full-wave liquid crystal retarder (Thorlabs, LCC1113-MIR, LCC25) for light manipulation before reaching the sample. The sample's image plane is aligned with the focal plane of an imaging lens set, comprising a 4 mm aspheric lens and a 25 mm lens. The transmitted light is captured and visualized using a high-resolution 640 × 512 pixels home-made MCT focal array, post-analyzer. Notably, the MWIR camera is cooled to approximately 80 K for optimal measurement performance. In linear polarization experiments, the full-wave liquid crystal retarder is intentionally omitted, reflecting a specific experimental choice. This precise and controlled setup underlines the reliability of our measurements in the mid-infrared range.

### Metasurface design and simulation
For designing metaatoms, we employ the three-dimensional finite difference time domain (FDTD) method, provided by Lumerical Inc., to calculate transmittance and phase distributions. The holograms are constructed using Python v3.8.13 and PyTorch v1.12.1 (Facebook Inc.), with the vectorial diffraction neural network (VDNN) formulated within this environment. The Adam optimizer in PyTorch is used as the automatic differentiation library, essential for holographic design optimization, while Numpy v1.21.5 enhances data processing efficiency. The angular spectrum diffraction algorithm is incorporated for computational efficiency in diffraction calculations. For arbitrary input polarization state, neural network models were trained with a fixed learning rate of 0.015. The number of iterations varied depending on the model's complexity. For instance, the linear, elliptic, and circular polarization models underwent 300 iterations each due to their smaller sizes, while the three-wavelength nine-channel model underwent 800 iterations, and the 55-channel model underwent 1500 iterations. To train the vectorial diffraction neural network, a workstation was specifically configured with two NVIDIA GeForce RTX 3090 GPUs (Nvidia Inc.), an AMD EPYC 7513 32-Core Processor (AMD Inc.), and 256 GB of RAM, running the Windows 10 operating system (Microsoft Inc.). The training time for a diffractive model with 55 polarization channels was approximately 0.5 h.

For large array hologram simulations (800 × 800 metaatoms), we utilize the vector integration algorithm based on Rayleigh-Sommerfeld diffraction theory, chosen to overcome the computational limitations of FDTD software, allowing for practical simulation of extensive metaatom arrays.

## Data availability
Relevant data supporting the key findings of this study are available in the article and Supplementary Information file. All raw data generated in this study are available from the corresponding authors upon reasonable request.

## Code availability
The code used for data analysis during this study is available upon reasonable request from the corresponding authors.

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

## Acknowledgements

This work was supported by Strategic Priority Research Program (B) of Chinese Academy of Sciences (XDB0580000, GJ0090406, XDB43010200); National Key Research and Development Program of China (2023YFA1406900); National Natural Science Foundation of China (62222514, 62350073, U2341226, 61991440, 62204249, 62305363); Shanghai Science and Technology Committee (23ZR1482000, 22JC1402900, 21ZR1402200); Natural Science Foundation of Zhejiang Province (LR22F050004); Shanghai Municipal Science and Technology Major Project (2019SHZDZX01); Youth Innovation Promotion Association (Y2021070) and International Partnership Program (112GJHZ2022002FN) of Chinese Academy of Sciences; Shanghai Human Resources and Social Security Bureau (2022670), Fundamental Research Funds for the Central Universities (2232022A-11) and China Postdoctoral Science Foundation (2023T160661, 2022TQ0353, 2022M713261). This work was partially carried out at the Center for Micro and Nanoscale Research and Fabrication in University of Science and Technology of China and Soft Matter Nanofab (SMN180827) in ShanghaiTech University.

## Author contributions

G.H.L. conceived the idea. J.W., J.C. and F.L.Y. performed the theoretical calculation and numerical simulation. G.H.L. and J.C. fabricated the sample. J.W., J.C., F.L.Y., J.X.W. and R.S.C. built the optical platform and conducted metasurface experiments. J.W., Z.Y.Z. and X.N.L analyzed the data. G.H.L., J.W., J.C., F.L.Y. and H.Z.X. prepared the manuscript with input from all authors. G.H.L., X.S.C. and W.L. initialized and supervised the project.

## Competing interests

The authors declare no competing interests.
