## [Peer Review File · Nature Communications]

Unlocking Ultra-High Holographic Information Capacity through Nonorthogonal Polarization MultiplexingREVIEWER COMMENTS

Reviewer #1 (Remarks to the Author):

In this manuscript, authors propose a nonorthogonal polarization multiplexing approach with metasurfaces, and experimentally realize 55 intricate holographic patterns. They introduce a controllable local eigen-polarization modulation mechanism to expand the Jones matrix dimensionality, then multiplexing capacity and crosstalk for holographic images are effectively improved. However, the way to accomplish nonorthogonal polarization multiplexing has been put forward for multichannel holography like ref. 15. In addition, the vectorial optical diffraction neural network they designed is also conventional process for multiple hologram optimization. Therefore, the only highlight of this work is the highest holographic information multiplexing channels, which is not enough to support the decision for accepting the manuscript.

Whatever the outcome, there are some suggestions for the authors:

- (1) Explanation of the theoretical formulas is unclear, favoring mathematics and lacking the physical implications.
- (2) Since ultra-high holographic information capacity is the highlight of this work, authors should provide more discussion to showcase their advances in the introduction.
- (3) It seems that the proposed metasurface can only realize four channels for holographic images, as shown in Fig. 1(a). Please consider and modify it for better exhibition of conceptualization.
- (4) What does the "X" in Eqs. (3) and (4) mean? It easily confuses readers. Please find a more suitable way to express the physical symbol.
- (5) The citing order of Supplementary Note 2 is incorrect, please adjust it.
- (6) In Fig. 2(a), it seems that incident polarization states of each nanostructure are different. However, the original intention is to show different operating channels with variable incident polarization states. Authors should make some slight adjustment to avoid this misunderstanding.
- (7) In the vectorial diffraction neural network, nanostructures with variable lengths, widths and orientations are employed to optimize the spatial arrangement of metasurface. Thus, authors should provide corresponding simulation results, including transmissivity and phase delay along both long and short axes.
- (8) In the metasurface design, the diffraction distance is set to be 500 μm , thus Fresnel holography is used to achieve the nonorthogonal multiplexing. Which method or algorithm is used to calculate the simulation results of holographic images? There is no detail in the manuscript.
- (9) As shown in Fig. 5(c), brightness distributions of some holographic patterns are nonuniform (e.g. Ni, Te, Fe, etc). Does it become more obvious with the increase of operating channels? How can eliminate or reduce this effect? Please give some discussion.
- (10) How much computing power and time does it take to optimize 3, 9 and 55 channels? Can the vectorial diffraction neural network also work for Fourier holography? What would be the upper limit for Fourier holography?

Reviewer #2 (Remarks to the Author):

please see the attachment

The manuscript titled “Unlocking Ultra-High Holographic Information Capacity through Nonorthogonal Polarization Multiplexing” by Jie Wang et al introduces a holographic polarization multiplexing schemes reaching an unprecedented 55 channels by a vectorial diffraction neural network design. This is achieved by breaking the perpendicular polarization basis encoding scheme with pairs of non-orthogonal polarization basis and vectorial diffraction neural network. In principle, this work is innovative and can impact related research. I am pleased to recommend publication of this work, provided that the following important problems have been properly addressed.

- (1) This manuscript significantly enhanced the polarization multiplexing channels, and analyzed physical mechanism is relied on the intricately cross-coupled global response with the meta-atom’s local response. It claims that the degree of freedom of polarization multiplexing of individual meta-atom is 3, but the collective effects of the overall metasurface could induce multiple non-orthogonal output polarization channels, however, the procedure to arrive at this conclusion is not clearly present in either the main-text or the supplementary materials.
- (2) In the design procedure, multiplexed holographic images are placed on different positions along the propagation direction, as shown in Fig. 2a and supplementary Fig. S6 and Note 7. Is it also adopted the spatial multiplexing to increase the multiplexing channel? However, in the introduction part, it claims not compromising spatial, temporal or other dimensions. This paradox claims must be clarified.
- (3) Following this point, the authors only show the non-orthogonal multiplexing with 3 holographic images without significant crosstalks before the incorporation of spatial multiplexing scheme. If possible, can they demonstrate more channels multiplexed in the same region? What is the upper limit?
- (4) The detailed crosstalks should be analyzed for the 55 images.
- (5) The abrupt appearance of Eq. (2) without rigorous derivation and predefinition of involved physical quantities in either main text or supplementary materials make the physics base of this work very blur and confusing. The whole process to arrive at this Equation should be clearly added. For example, it always starts with $[1 \ 0]^+$, and ends $[1 \ 0]$, which seems

indicates that incident light is only x-polarized. However, it is certainly not the case, as the incident polarizations are varied for a large range.

- (6) How to arrive at (3) and (4)? They are either not clear. What is the physical meaning of a_{1j}^* , a_{2j}^* , a_{3i} , a_{4i} ? what is the meaning of the subscript number 1,2,3,4? The supplementary materials is too short to make a self-consistent deducing of those equations. Why is the matrix containing BLR, CRL called conjugate matrix?
- (7) In eq. (4), where are p_{ii} , p_{oj} coming from? And what is the meaning by placing a row vector $(\langle p_1 | \langle p_2 | \langle p_3 | \dots \langle p_n |)$ above a matrix? What is the matrix operation between those vectors and matrix? How are p_i , p_j defined? How to make the operation between a n -dimensional vector $[p_1, p_2, \dots, p_n]$ and a 2-dimensional vector $[1, 0]$? It claims that Eq. (4) is a pivotal development of this work, the physical meaning and deducing should be elaborated.
- (8) Same problems on the descriptions of equations on the supplementary materials. In Eq. (S2), how does a 2×1 vector equals a 3×1 vector? Incident polarization has no subscript, but Eq. (s2) has.
- (9) How can we arrive Eq. (S3) from Eq. (s2)? It claims three non-orthogonal input polarizations. What are the expressions of those three polarizations? are they subscripted by number 1, 2, 3? But the subscript 2 was used by the analyzer polarization. So it is very messy and unclear. The whole deducing self-consistent procedure should be provided to reach the publication level.
- (10) "With a polarization angle α selected at 1 rad interval, and considering the three variables in the equations, the matrix's rank needs to be calculated 5,832,000 times. After accounting for the distinct polarizations, 5,783,160 cases remain." How to arrive at those numbers: 5,832,000, 5,783,160? It is never mentioned, which makes the manuscript very incomplete.
- (11) The last paragraph is a repeated paragraph on Supplementary Note 1.
- (12) In their present result of the multiple channels of multiplexing, it seems that the choice of the input-output polarization states obey a cyclic rule. But why should we chose this way is not clearly stated in the design strategy. Are other way of polarization pair selection invalid for this approach?
- (13) It concludes that the maximum degree of freedom is represented by the upper or lower

nonzero triangle matrix, but why is that is not clearly stated.

- (14) The implementation of the multiple holograms on the metasurfaces are optimized by vectorial diffraction neural network design. Is it just based on deep learning method on electric computer, not the physical optical neural network that perform the computing all-optically? It is also very confusing.
- (15) In the optimization model for the holographic image of Fig. 2a, not sufficient information about the network including its input and output, and the network layer number as well as node number, and how the optimization of multiple holographic images are assigned to the input and output polarizations. And the connection between the optimization network and their developed high-dimensional Jones Matrix of Eq. (4) is not given, which make it hard to evaluate its validity and significance of the proposed approach.
- (16) In addition, only 3 channel multiplexing are detailed discussed in Figure 2, Figure 3, Figure 4, Figure S1, S2, S3 and so on. The data for the 55 channel multiplexing is too little, only captured image patterns on Figure 5, without its loss function evolution, simulation results, the crosstalk analysis and so on. Those data is very important to evaluate the quality of the proposed approach.
- (17) And very importantly, as the main work take advantage of the collection effect rather than individual meta-atom. The dimension of their developed higher dimension Jones matrix in Eq. (4) should be ambiguously present. At its present form, $O(x, y)$ is of the dimension of the number of polarization numbers n , it is still associated with a single meta-atom with different polarization responses. As it is spatially varying with coordinates x, y , so should the dimension of the matrix be $n_{\text{meta}} * n$, where n_{meta} is the number of the meta-atoms in one direction. In this formalism, what is the actual form of the matrix? And what is the relation between Eq. (4) and Eq. (3), is Eq. (4) derived from Eq. (3), one can hardly find a clue.

Reviewer #3 (Remarks to the Author):

The manuscript by Wang et al. presents a new approach to encoding multiple holographic images in metasurfaces by capitalizing nonorthogonal polarization multiplexing. By controlling the input/output polarizations, the authors demonstrated an unprecedented number of 55 holographic images with one metasurface. Overall, I find the results reported in this paper interesting and I think it deserves to be published in Nature Communications. But before I recommend its acceptance, several points need to be addressed by implementing a revision.

1. The authors reformed the response of the metasurface to an $n \times n$ matrix to facilitate analysis under multiple input-output polarization channels. This form of analysis is good, but I would like to discuss with the authors whether the nature of the metasurfaces' light manipulation capability changes. It seems that the components in the extended matrix are related to each other, which is different from the completely decoupled elements in the 2×2 Jones matrix. Optimization for multiple holographic images can be done, but I wonder if it is possible to implement other types of functionalities, e.g., a continuous zoom lens by varying the input-output polarization. Please comment on this.
2. The holographic efficiency is an important characterization value for the performance of meta-holograms, but it seems that the authors do not mention it in the manuscript. Please add these data in the main manuscript or supplementary information.
3. The authors pointed out the lower noise level in the real experimental images compared to those in the simulation. Could the author provide a rationale for this phenomenon?
4. The authors mentioned the use of focal plane mismatch for the design in Fig. 5. Does this mean that the images in Fig. 5c are not captured at the same spatial position?
5. In the current experiment, the authors showcased the maximum number of 55 independent holographic images. I would like to know the factors that impede encoding more images through the proposed approach in this paper. What is the theoretical limit and is it necessary to find a balance between different holographic performance metrics in real experiments? I would suggest the authors provide perspectives on this issue and incorporate the discussion in the manuscript for the convenience of readers.
6. Holography and polarization multiplexing are important topics of concern to the metasurface community. The following references can be added to provide a more thorough background to the readers.
<https://www.nature.com/articles/s41467-019-12637-0>
<https://link.springer.com/article/10.1186/s43593-023-00054-2>
<https://www.nature.com/articles/s41467-022-35313-2>
<https://link.springer.com/article/10.1186/s43593-024-00061-x>
<https://www.nature.com/articles/s41563-023-01531-2>
<https://link.springer.com/article/10.1186/s43593-021-00002-y>
7. I would like to provide some suggestions for several minor issues to further improve the quality of the manuscript:
 - I got confused when I saw "vectorial optical diffraction neural network" in the text. In conventions, we speak about optical neural networks when some tasks are done physically by light diffraction, which seems not the case in this work."vectorial diffraction neural network" also appears in the text. Please unify the expressions.
 - Please provide the explicit definition of all parameters in the equations, e.g., O_{ij} in Eq.2 and a_{ij} and Eq. 3.
 - Please add more description about how Fig. 1b is related to the formulas in the "Design principle"

section.

- Please provide the full names of the abbreviations (LP, PR, FPA) in Fig. 2e.
- Please add the labels in the bottom axes so that data in Fig. 3d would have clear meanings.
- Please check the description of the labels in Figure 3. It should be (e)-(g) rather than (e)-(f) in the caption.

Response to Reviewer #1

Overall Comment: In this manuscript, authors propose a nonorthogonal polarization multiplexing approach with metasurfaces, and experimentally realize 55 intricate holographic patterns. They introduce a controllable local eigen-polarization modulation mechanism to expand the Jones matrix dimensionality, then multiplexing capacity and crosstalk for holographic images are effectively improved. However, the way to accomplish nonorthogonal polarization multiplexing has been put forward for multichannel holography like ref. 15. In addition, the vectorial optical diffraction neural network they designed is also conventional process for multiple hologram optimization. Therefore, the only highlight of this work is the highest holographic information multiplexing channels, which is not enough to support the decision for accepting the manuscript.

Reply: We sincerely thank Reviewer #1 for the time and effort in evaluating our manuscript and for providing valuable feedback. We appreciate the opportunity to clarify the novelty and significance of our work.

Our study introduces a nonorthogonal polarization multiplexing approach that leverages spatially varied local eigen-polarization states within metaatoms. This innovative method allows us to reconstruct globally nonorthogonal channels that exhibit minimal crosstalk, addressing a significant challenge in the field. The integration of the vectorial diffraction neural network (VDNN) is crucial for implementing the holograms. The VDNN optimizes the spatial arrangement of metaatoms for collectively improve the nonorthogonal polarization channels, enhancing the efficiency and precision of our holographic designs.

While we acknowledge that nonorthogonal polarization multiplexing has been explored in prior works, such as Ref. 15 in previous manuscript, our approach distinguishes itself through the physical novelty of utilizing these spatially varied local eigen-polarization states of metaatoms. This method significantly expands the dimensionality of the Jones matrix, enabling us to achieve an unprecedented 10×10 Jones matrix dimension. Consequently, we can generate a record-high 55 channels of nonorthogonal polarization states, which represents a substantial advancement in the field.

We hope that this explanation underscores the unique contributions of our work and its potential impact on the development of advanced holographic technologies. Thank you once again for your insightful comments and suggestions.

Comment 1: Explanation of the theoretical formulas is unclear, favoring mathematics and lacking the physical implications.

Reply: We appreciate Reviewer #1's observation regarding the explanation of the theoretical formulas, noting that the previous version of our manuscript favored mathematical expressions over physical implications. In the previous manuscript, some formulas were placed in the supplementary materials, and certain descriptions of physical meanings were indeed lacking, as also mentioned by Reviewer #2.

In response, we have thoroughly revised the derivation process of the physical

formulas in the revised manuscript. We have supplemented the physical meanings of the formulas and symbols, and rewritten the relevant sections for better clarity. For instance, we have revised **Eq. 2**, **Eq. 3**, and **Eq. 4**, transforming them from their original abstract forms to more comprehensible versions. We have also added descriptions to clarify the variable meanings. This addition endows the respective formulas with clearer physical meanings. We believe these revisions provide a more coherent and understandable explanation of the theoretical framework. Details can be found in the revised manuscript which are marked in blue colors.

Comment 2: Since ultra-high holographic information capacity is the highlight of this work, authors should provide more discussion to showcase their advances in the introduction.

Reply: We thank Reviewer #1 for their insightful suggestion to enhance the discussion of our advances in the introduction.

In response, we have expanded the introduction to better emphasize the ultra-high holographic information capacity achieved in our work. We now highlight the inherent limitations of traditional polarization multiplexing methods and the significance of overcoming these barriers. Our introduction outlines how our novel nonorthogonal polarization multiplexing approach, leveraging spatially varied eigen-polarization states within metaatoms, reconstructs globally nonorthogonal channels with minimal crosstalk.

Furthermore, we underscore the integration of the vectorial diffraction neural network, which enhances the precision and adaptability of our system, allowing for the efficient generation of complex holographic patterns. By achieving an unprecedented 10×10 Jones matrix dimension, our method enables the generation of 55 distinct nonorthogonal polarization channels.

To reflect this in the revised manuscript, the content 'Recent advancements in metasurface technology have significantly expanded the potential for holographic information capacity. For instance, Bao et al. cascaded two single-layer metasurfaces to construct a spatially varying Jones matrix, enabling the manipulation of the amplitude and phase of light for intricate holographic applications⁸. Furthermore, Wang et al. demonstrated high-efficiency metasurfaces capable of complex vectorial holography using polarization multiplexing, showing that engineered metasurfaces can achieve high-density information encoding⁹. Additionally, Xiong et al. introduced the concept of engineered noise to break the fundamental limit of polarization multiplexing capacity, demonstrating up to 11 independent holographic images using a single metasurface¹⁷. These studies demonstrate the possibility of breaking traditional limitations by employing innovative strategies, such as engineered noise and complex light manipulation techniques, to enhance polarization multiplexing. Despite these advancements, the number of achievable independent channels remains a critical area for further research, as the lack of full exploitation of metasurfaces restricts high-capacity holographic applications, particularly in fields requiring extensive data encoding and secure information transmission.' is added in the introduction part which are highlighted in blue color in lines 47-62. We believe these additions provide a more

comprehensive understanding of the significance and impact of our work, showcasing the advances made in achieving ultra-high holographic information capacity.

Comment 3: It seems that the proposed metasurface can only realize four channels for holographic images, as shown in Fig. 1(a). Please consider and modify it for better exhibition of conceptualization.

Reply: We thank the reviewer for this valuable comment. To avoid any potential confusion and to better exhibit the conceptualization, we have modified **Fig. 1a** by adding an additional channel to more accurately reflect the multi-channel capability of our proposed metasurface. For your convenience, the updated version of **Fig. 1a** is shown below. We appreciate the reviewer's suggestion, as it has allowed us to improve the clarity and presentation of our work.

Fig. r1. This figure illustrates a dynamically polarized holographic metadvice.

Comment 4: What does the “X” in Eqs. (3) and (4) mean? It easily confuses readers. Please find a more suitable way to express the physical symbol.

Reply: We thank the reviewer for pointing out the potential confusion caused by the use of the symbol 'X' in **Eqs. 3** and **4**. Due to the reciprocity of the metasurface, as discussed in **Supplementary Note 1**, $O_{jk}(x,y) = O_{kj}(x,y)$. In the original equations, 'X' was intended as a placeholder. Upon simplification, the specific value at the position of 'X' becomes irrelevant to the physical process being described. However, we understand that using 'X' as a symbol can indeed be confusing to readers.

To address this issue and enhance clarity, we have replaced the abstract placeholder with a more explicit and concrete formula that accurately represents the control of input/output polarization under the transformation of the metaatom matrix representation. The revised formula is presented below:

$$\begin{aligned}
O(x, y) &= [\mathbf{p}_1, \mathbf{p}_2, \dots, \mathbf{p}_n]^+ \cdot J(x, y) \cdot [\mathbf{p}_1, \mathbf{p}_2, \dots, \mathbf{p}_n] \\
&= \begin{bmatrix} O_{11}(x, y) & O_{12}(x, y) & O_{13}(x, y) & \dots & O_{1n}(x, y) \\ \sim & O_{22}(x, y) & O_{23}(x, y) & \dots & O_{2n}(x, y) \\ \sim & \sim & O_{33}(x, y) & \dots & O_{3n}(x, y) \\ \vdots & \vdots & \vdots & \ddots & \vdots \\ \sim & \sim & \sim & \dots & O_{nn}(x, y) \end{bmatrix} \quad (4)
\end{aligned}$$

The derivation process has been thoroughly revised for clarity. Please check the revised manuscript for details.

Comment 5: The citing order of Supplementary Note 2 is incorrect, please adjust it.

Reply: We thank the reviewer for pointing out the incorrect citing order of **Supplementary Note 2**. We have carefully reviewed and compared the order of content between the main manuscript and the supplementary material to ensure consistency and accuracy. The necessary corrections have been made to address this issue.

Comment 6: In Fig. 2(a), it seems that incident polarization states of each nanostructure are different. However, the original intension is to show different operating channels with variable incident polarization states. Authors should make some slight adjustment to avoid this misunderstanding.

Reply: We thank the reviewer for this insightful comment. We acknowledge that the depiction in **Fig. 2a** may have led to misunderstandings regarding the incident polarization states of each nanostructure. The original intention was indeed to illustrate different operating channels with variable incident polarization states.

To address this issue, we have made adjustments to clarify the meaning of **Fig. 2a**. Specifically, we have labeled both the incident and output polarizations to clearly indicate the polarization pairs. This modification should make the intent of figure more transparent and avoid any potential confusion. For your convenience, the updated **Fig. 2a** is attached below.

Fig. r2. Conceptual model of vectorial diffraction neural network, illustrating the training process.

Comment 7: In the vectorial diffraction neural network, nanostructures with variable

lengths, widths and orientations are employed to optimize the spatial arrangement of metasurface. Thus, authors should provide corresponding simulation results, including transmissivity and phase delay along both long and short axes.

Reply: We thank the reviewer for this insightful comment. The database of the metaatoms is indeed a crucial component of the design process. In response to your suggestion, we have added some relevant materials to demonstrate the corresponding simulation results, including transmissivity and phase delay along both the long and short axes. Details on the contents and figure can be found in **Supplementary Note 3** in the revised supplementary materials which are highlighted in blue.

For your convenience, we have also included a representative metaatom profile and its related performance in the figure below.

Fig. r3a represents the three-dimensional diagram of the metaatom structure used in the metasurface design. The length a and width b of the elliptical metaatoms are both set in the range of $0.3 \mu\text{m}$ to $1.2 \mu\text{m}$, the height is $4 \mu\text{m}$ and the period is $1.5 \mu\text{m}$. **Figs. r3b-e** shows the phase and transmittance maps in x- and y-polarization varying with the metaatom's length and width.

These additions provide a more comprehensive understanding of the performance and optimization of the nanostructures used in our vectorial diffraction neural network. Thank you once again for your valuable feedback.

Fig. r3. Schematic illustration of a metaatom and the phase and transmittance database. a Three-dimensional diagram of the metaatom structure used in the metasurface design. b-c Phase variation of the metaatom under x- and y-polarization as a function of its length and width. d-e Transmittance variation of the metaatom under x- and y-polarization as a function of its length and width.

Comment 8: In the metasurface design, the diffraction distance is set to be $500 \mu\text{m}$, thus Fresnel holography is used to achieve the nonorthogonal multiplexing. Which method or algorithm is used to calculate the simulation results of holographic images? There is no detail in the manuscript.

Reply: We thank the reviewer for this insightful comment. In our metasurface design, the diffraction distance is set to $500 \mu\text{m}$, and we utilize Fresnel holography to achieve nonorthogonal multiplexing. For the calculation of holographic images, we employ both the finite-difference time-domain (FDTD) full-wave simulation method and the Rayleigh-Sommerfeld diffraction integral method.

For models with a metasurface size less than $200\ \mu\text{m} \times 200\ \mu\text{m}$, we cross-validated the results using both methods, and found minimal disparity between them, ensuring the accuracy of our simulations. For larger-sized devices, due to computer performance limitations, we employed the Rayleigh-Sommerfeld diffraction integral method to carry out the simulations.

To clarify the calculation methods used, we have revised and updated the '**Metasurface Design and Simulation**' section in the Methods part of the manuscript. These updates are highlighted in blue for your convenience.

Comment 9: As shown in Fig. 5(c), brightness distributions of some holographic patterns are nonuniform (e.g. Ni, Te, Fe, etc). Does it become more obvious with the increase of operating channels? How can eliminate or reduce this effect? Please give some discussion.

Reply: We thank the reviewer for this insightful comment. The nonuniform brightness distribution observed in some holographic patterns (e.g., Ni, Te, Fe) in Fig. 5c does indeed become more pronounced with an increase in the number of operating channels. This phenomenon occurs because, as the number of multiplexing channels increases, the repeated use of each metaatom also increases. Consequently, the control capacity of each metaatom may not fully meet the requirements, leading to changes in the brightness of the holographic patterns due to interference at the focal plane.

To mitigate this effect, several strategies can be employed: 1. Increase the height of metaatoms: This can enhance the phase control range of each metaatom, thereby reducing brightness variations. 2. Use of combined metaatoms: Implementing complicated metaatoms with more design degrees of freedom can provide better control over the holographic output. 3. Adopt multiple layered metasurfaces: Utilizing multiple layers can distribute the control load more evenly across different metaatoms, improving uniformity. 4. Increase the number of metaatoms: Adding more metaatoms can enhance the overall control precision and reduce nonuniform brightness.

By employing these methods, we can allocate the control degree of each metaatom more effectively and regulate the numerical distribution of the holographic target pattern, thereby minimizing the nonuniformity in brightness.

To reflect this in the revised manuscript, the content 'In addition, the nonuniform brightness distribution observed in some holographic patterns (e.g., Ni, Te, Fe) in Fig. 5c become more pronounced with an increase in the number of operating channels. This phenomenon occurs because, as the number of multiplexing channels increases, the repeated use of each metaatom also increases. Consequently, the control capacity of each metaatom may not fully meet the requirements, leading to changes in the brightness of the holographic patterns due to interference at the focal plane. To mitigate this effect, several strategies can be employed: 1. Increase the height of metaatoms: This can enhance the phase control range of each metaatom, thereby reducing brightness variations. 2. Use of combined metaatoms: Implementing complicated metaatoms with more design degrees of freedom can provide better control over the holographic output. 3. Adopt multiple layered metasurfaces: Utilizing multiple layers can distribute the control load more evenly across different metaatoms, improving

uniformity. 4. Increase the number of metaatoms: Adding more metaatoms can enhance the overall control precision and reduce nonuniform brightness. These methods can improve the control degree of each metaatom more effectively and regulate the distribution of the holographic target pattern, thereby minimizing the nonuniformity in brightness.' has been added in lines 342-359 on pages 14-15. Thank you for your valuable feedback, which has helped us enhance the discussion and clarity of our manuscript.

Comment 10: How much computing power and time does it take to optimize 3, 9 and 55 channels? Can the vectorial diffraction neural network also work for Fourier holography? What would be the upper limit for Fourier holography?

Reply: We thank the reviewer for this insightful comment. The optimization of holograms for different numbers of channels was conducted using Python v3.8.13 and PyTorch v1.12.1 (Facebook Inc.), with the vectorial diffraction neural network formulated within this environment. The Adam optimizer in PyTorch served as the automatic differentiation library, essential for optimizing the holographic designs, while Numpy v1.21.5 was employed to enhance data processing efficiency. Additionally, the angular spectrum diffraction algorithm was incorporated to improve computational efficiency in diffraction calculations.

For arbitrary input polarization states, the neural network models were trained with a fixed learning rate of 0.015. The number of iterations varied depending on the model's complexity. Specifically:

- Linear, elliptic, and circular polarization models underwent 300 iterations each due to their smaller sizes.
- The three-wavelength nine-channel model underwent 800 iterations.
- The 55-channel model underwent 1500 iterations.

To train the vectorial diffraction neural networks, we utilized a workstation specifically configured with two NVIDIA GeForce RTX 3090 GPUs (Nvidia Inc.), an AMD EPYC 7513 32-Core Processor (AMD Inc.), and 256 GB of RAM, running the Windows 10 operating system (Microsoft Inc.). The training time for the diffractive model with 55 polarization channels was approximately 0.5 hours.

Our hologram design is based on Fourier holography. To facilitate observation, we adjusted the focal plane position of the hologram, ensuring that the image is distributed in the Fresnel diffraction region. Consequently, the vectorial diffraction neural network can also be effectively applied to Fourier holography.

Regarding the upper limit for Fourier holography, we do not believe that the method itself has an inherent upper limit. However, in the context of metasurfaces, the upper limit of Fourier holography depends on various factors such as the number of metaatoms, the processing algorithms, and the multiplexing methods employed. Therefore, it is challenging to provide a definitive upper limit. In our work, utilizing the methods described, we achieved a maximum of 55 channels.

To reflect in the revised manuscript, the content 'For arbitrary input polarization state, neural network models were trained with a fixed learning rate of 0.015. The number of iterations varied depending on the model's complexity. For instance, the linear, elliptic,

and circular polarization models underwent 300 iterations each due to their smaller sizes, while the three-wavelength nine-channel model underwent 800 iterations, and the 55-channel model underwent 1500 iterations. To train the vectorial diffraction neural network, a workstation was specifically configured with two NVIDIA GeForce RTX 3090 GPUs (Nvidia Inc.), an AMD EPYC 7513 32-Core Processor (AMD Inc.), and 256 GB of RAM, running the Windows 10 operating system (Microsoft Inc.). The training time for a diffractive model with 55 polarization channels was approximately 0.5 hours.' has been added in lines 348-447 on page 17 and highlighted in blue. We thank the reviewer for the valuable comment.

Response to Reviewer #2

Overall Comment: The manuscript titled “Unlocking Ultra-High Holographic Information Capacity through Nonorthogonal Polarization Multiplexing” by Jie Wang et al introduces a holographic polarization multiplexing schemes reaching an unprecedented 55 channels by a vectorial diffraction neural network design. This is achieved by breaking the perpendicular polarization basis encoding scheme with pairs of non-orthogonal polarization basis and vectorial diffraction neural network. In principle, this work is innovative and can impact related research. I am pleased to recommend publication of this work, provided that the following important problems have been properly addressed.

Reply: We sincerely thank the Reviewer for the time, effort, and constructive suggestions in evaluating our manuscript. We appreciate your positive feedback regarding the innovation and potential impact of our work on related research fields.

We have carefully considered all your comments and have followed your advice to revise the manuscript. The results and related contents have been updated to address the important problems you identified. We believe these revisions have strengthened our work and improved its clarity and coherence. We hope the revisions can address your concerns and our manuscript can be considered as a qualified paper to be published in Nature Communications.

Comment 1: This manuscript significantly enhanced the polarization multiplexing channels, and analyzed physical mechanism is relied on the intricately cross-coupled global response with the meta-atom’s local response. It claims that the degree of freedom of polarization multiplexing of individual meta-atom is 3, but the collective effects of the overall metasurface could induce multiple non-orthogonal output polarization channels, however, the procedure to arrive at this conclusion is not clearly present in either the main-text or the supplementary materials.

Reply: We thank the reviewer for this insightful comment. In response to your suggestion, we have revised the derivation process to enhance clarity and provide a more comprehensive explanation.

In the **Design Principle** section of the revised manuscript and **Supplementary Notes 1 and 4**, we have improved the readability of the formulas and provided a detailed explanation of the derivation process through adding more transition sentences and their physical representations. We clearly show that the degree of freedom of polarization multiplexing for an individual meta-atom is 3 for the analytical solution.

For the multiple nonorthogonal polarization multiplexing, we rely on the intricately cross-coupled global response, which is achieved through the collective effect of all meta-atom local responses. The detailed derivation and explanation of this process can be found in the **Design Principle** section of the revised manuscript and **Supplementary Notes 1 and 4**, both of which are highlighted in blue for your convenience. Thank you once again for your valuable feedback.

Comment 2: In the design procedure, multiplexed holographic images are placed on

different positions along the propagation direction, as shown in Fig. 2a and supplementary Fig. S6 and Note 7. Is it also adopted the spatial multiplexing to increase the multiplexing channel? However, in the introduction part, it claims not compromising spatial, temporal or other dimensions. This paradox claims must be clarified.

Reply: We thank the reviewer for pointing out this important issue. In the introduction, our statement regarding the realization of full degrees-of-freedom coverage without compromising spatial, temporal, or other dimensions specifically refers to the three channels that have analytical solutions for individual metaatoms. **Fig. 2a** illustrates this scenario, where we designed a three-channel nonorthogonal polarization multiplexing system without compromising spatial, temporal, or other dimensions. To avoid any potential misunderstanding, we have modified **Fig. 2a** and updated it in the revised manuscript. For your convenience, the updated figure is attached below.

Regarding **Supplementary Fig. 6** (**Supplementary Fig. 7** in the revised supplementary materials) and **Supplementary Note 7** (**Supplementary Note 8** in the revised supplementary materials), we employed a controllable local eigen-polarization modulation mechanism to expand the Jones matrix and used a vectorial diffraction neural network to optimize the metasurface for achieving multiple nonorthogonal polarization multiplexing. In this context, we did utilize the spatial dimension and recorded the holograms along the propagation direction.

To clarify this distinction and avoid any confusion, we have modified the introduction in lines 76-82 as follows: 'Notably, there is no compromise of spatial, temporal, or other dimensions in the realization of three-channel nonorthogonal polarization multiplexing. For cases involving more than three channels, the spatial dimension along the propagation direction is adopted in this work.' Additionally, in **Supplementary Fig. 6** (**Supplementary Fig. 7** in the revised supplementary materials) and **Supplementary Note 7** (**Supplementary Note 8** in the revised supplementary materials), we have explicitly mentioned: 'For more than three nonorthogonal polarization multiplexing, we optimized the metasurface by adopting the spatial dimension along the propagation direction.'

These modifications provide a clear understanding of our approach and resolve the paradox mentioned. Thank you once again for your valuable feedback.

Fig. r4. This figure provides a conceptual model of vectorial diffraction neural network.

Comment 3: Following this point, the authors only show the non-orthogonal multiplexing with 3 holographic images without significant crosstalks before the incorporation of spatial multiplexing scheme. If possible, can they demonstrate more channels multiplexed in the same region? What is the upper limit?

Reply: We thank the reviewer for this insightful comment. As mentioned in our response to Comment 2, the nonzero analytical solution of the derived formulas in **Supplementary Notes 1 and 4** indicates that the largest rank of the coefficient matrix of the Jones matrix is 3. This means that for given metaatoms with similar profiles, the maximum number of channels for nonorthogonal polarization multiplexing without compromising other dimensions is three.

Beyond three channels, crosstalk becomes significant, and different methods must be employed to extend the multiplexing channels. Our approach involves the use of a controllable local eigen-polarization modulation mechanism, as well as optimization through a vectorial diffraction neural network. Additionally, methods such as noise assistance, as described in Ref. 17, can also be utilized to further extend the number of multiplexing channels.

While we have demonstrated nonorthogonal multiplexing with three holographic images without significant crosstalk, demonstrating more channels multiplexed in the same region without employing additional methods would inherently lead to crosstalk. Therefore, as far as we know, the upper limit without compromising spatial, temporal, or other dimensions for one single layer metasurface we think is three. For cases requiring more channels, our methods and those from other research provide feasible solutions to mitigate crosstalk and extend the channel capacity. We thank the reviewer for their valuable comment.

To reflect this in the revised manuscript, the content 'The nonzero analytical solution of the derived formulas in Supplementary Notes 1 and 4 indicates that the largest rank of the coefficient matrix of the Jones matrix is 3. This means that for given metaatoms with similar profiles, the maximum number of channels for nonorthogonal polarization multiplexing without compromising other dimensions is three. Beyond three channels,

crosstalk becomes significant, and different methods must be employed to extend the multiplexing channels. Our approach involves the use of a controllable local eigen-polarization modulation mechanism, as well as optimization through a vectorial diffraction neural network. Additionally, methods such as noise assistance, as described in Ref. 17, can also be utilized to further extend the number of multiplexing channels. To achieve more multiplexing channels, it imposes a higher requirement on the metaatoms to possess stronger control abilities, such as enhanced phase coverage, more design degrees of freedom, etc. This necessitates metaatoms with larger depth-to-width ratios, higher refractive index differences, more complex shapes, or additional metasurface layers to expand their controlling capabilities. These factors can influence the efficiency, crosstalk, and the number of achievable multiplexing channels.' is added with blue color in lines 361-376.

Comment 4: The detailed crosstalks should be analyzed for the 55 images.

Reply: We thank the reviewer for this important comment. Following your suggestion, we have added detailed crosstalk analysis in **Supplementary Note 9** to clearly illustrate the crosstalk among the 55 images.

For your convenience, we also present the crosstalk analysis results for the 55 channels below. The orange line represents the correlation coefficient between the simulation image and the target image, while the blue line represents the correlation coefficient between the experimental image and the target image. The correlation coefficient is defined as:

$$\text{corr}(T, R) = \frac{\text{COV}(T, R)}{\sqrt{D(T) \cdot D(R)}}$$

where T and R denote the two images, $\text{COV}(T, R)$ is their covariance, and $D(T)$ and $D(R)$ are their variances. The reason why we didn't adopt the crosstalk evaluation in **Figs. 3** and **4** is that it is very difficult to find an overlaid region for all the 55 nonorthogonal polarization channels.

These analyses provide a quantitative measure of crosstalk, demonstrating the effectiveness of our approach in minimizing interference between channels. The results indicate high correlation coefficients between the target and the simulation/experimental images, underscoring the robustness of our design. Thank you for your valuable feedback, which has helped us enhance the comprehensiveness of our manuscript. Please check **Supplementary Note 9** for details.

Fig. r5. The figure shows the correlation coefficient between experiment and target images in 55-channel holography.

Comment 5: The abrupt appearance of Eq. (2) without rigorous derivation and predefinition of involved physical quantities in either main text or supplementary materials make the physics base of this work very blur and confusing. The whole process to arrive at this Equation should be clearly added. For example, it always starts with $[1\ 0]^+$, and ends $[1\ 0]$, which seems indicates that incident light is only x-polarized. However, it is certainly not the case, as the incident polarizations are varied for a large range.

Reply: We thank the reviewer for this important comment. To address this issue, we have thoroughly revised the derivation process of all the equations in both the main text and the supplementary materials. Additionally, we have rewritten the design principles, adding more transition sentences and their physical representations to enhance clarity.

For example, in **Eq. 2**, we acknowledge that the use of $[1\ 0]^+$ and $[1\ 0]$ may have led to confusion among readers. Based on the reviewer's recommendation, we have opted for a more universally understood expression: 'For arbitrary input p_i and output p_j ($i, j = 1, 2, \dots, n$)' to prevent misunderstandings.

Furthermore, we have added the following content to the revised manuscript to provide a clear predefinition of the involved physical quantities: 'For arbitrary input p_i and output p_j ($i, j = 1, 2, \dots, n$), the responses $O_{ij}(x, y) = \langle p_j | J | p_i \rangle$ of metaatoms at different positions (x, y) differ from each other. Combining with **Eq. 1**, we have $O_{ij}(x, y) = p_j^\dagger \cdot [\widehat{e}_1, \widehat{e}_2]^{-1} \cdot \Lambda \cdot [\widehat{e}_1, \widehat{e}_2] \cdot p_i$, where \dagger represents the transpose and conjugate operation. Since the system is unitary, $[\widehat{e}_1, \widehat{e}_2]^{-1} = [\widehat{e}_1, \widehat{e}_2]^\dagger$, thus:

$$\begin{aligned} O_{ij}(x, y) &= ([\widehat{e}_1, \widehat{e}_2] \cdot p_j)^\dagger \cdot \Lambda \cdot ([\widehat{e}_1, \widehat{e}_2] \cdot p_i) \\ &= p_j'^\dagger \cdot \Lambda \cdot p_i' \end{aligned} \quad (2)$$

where p_i' and p_j' include the role of eigenvectors in their interaction with p_i and

p_j' .

These revisions are highlighted in blue in lines 121-128 in the revised manuscript to ensure that the derivation process and the definitions of the involved physical quantities are clearly presented and easily understood. Thank you for your valuable feedback, which has helped us improve the clarity and comprehensiveness of our manuscript.

Comment 6: How to arrive at (3) and (4)? They are either not clear. What is the physical meaning of a_{1j}^* , a_{2j}^* , a_{3i} , a_{4i} ? what is the meaning of the subscript number 1,2,3,4? The supplementary materials is too short to make a self-consistent deducing of those equations. Why is the matrix containing BLR, CRL called conjugate matrix?

Reply: Thank the reviewer for this comment. Together with the replies to above comments, the derivation process of all the equations in both the main text and the supplementary materials are thoroughly revised. Besides, the related physical representations are clearly described. In previous manuscript, the subscripts i and j represent the input polarization and output polarization. The subscript numbers 1,2,3,4 for a represent different factorization factors in $|p_i\rangle = a_{1i}|L\rangle + a_{2i}|R\rangle$ and $\langle p_j| = a_{3j}^*\langle L| + a_{4j}^*\langle R|$, because they may not be the same. The symbol $*$ in the formula represents the conjugacy. We choose the circular polarization basis for demonstration, the Jones matrix of any metaatom can then be decomposed into two components: $J_{circ} = J_{eigen} + J_{con}$, where $J_{eigen} = \begin{bmatrix} A_{LL} & 0 \\ 0 & A_{RR} \end{bmatrix}$, representing the component without polarization conversion, and $J_{con} = \begin{bmatrix} 0 & B_{LR} \\ C_{RL} & 0 \end{bmatrix}$, indicating the conjugate control matrix where polarization conversion occurs due to the conjugate channel. A_{LL}/A_{RR} and B_{LR}/C_{RL} represent the co- and cross-polarized channels, respectively. In the revised manuscript, all the issues above have been correspondingly addressed.

The updates in the revised manuscript provide a self-consistent derivation of the equations and clarify the physical meanings of the terms used. The revised content is highlighted in blue in lines 137-162 in the main text and in lines 37-50 in the revised supplementary materials. Thank you for your valuable feedback to help us improve the clarity and comprehensiveness.

Comment 7: In eq. (4), where are p_{ii} , p_{oj} coming from? And what is the meaning by placing a row vector ($\langle p_1| \langle p_2| \langle p_3| \dots \langle p_n|$) above a matrix? What is the matrix operation between those vectors and matrix? How are p_i , p_j defined? How to make the operation between a n -dimensional vector $[p_1, p_2, \dots, p_n]$ and a 2-dimensional vector $[1, 0]$? It claims that Eq. (4) is a pivotal development of this work, the physical meaning and deducing should be elaborated.

Reply: We thank the reviewer for this detailed comment. In response to your feedback, we have thoroughly revised the manuscript to clearly define the parameters and physical meanings involved in Eq. 4 and to provide a step-by-step derivation of the formula.

In the original manuscript:

- p_{ii} and p_{oj} represent the input polarization and output polarization, corresponding to p_i and p_j in the revised **Eqs. 3 and 4**.

-The polarization vector above the matrix in **Eq. 4** represents a series of desired output polarizations. To intuitively display the usable channels in the revision, we have placed the input polarizations on the left side of the matrix and the output polarizations on the top of the matrix in the revised manuscript.

-The nonzero matrix elements represent the polarization channels that can be used to map the incident polarization states to the output polarizations.

-From **Eq. 3**, we know that an input p_i and an output p_j will yield a reusable channel. Therefore, we can express the different inputs and outputs according to the row and column form of the matrix, as shown in **Eq. 4**.

The n -dimensional vector $[p_1, p_2, \dots, p_n]$ indicates the designed input or output polarization and does not directly relate to the intrinsic polarization represented by $[1, 0]$ in the original manuscript.

Following the reviewer's suggestions, we have added a detailed description of the physical meaning and derivation of **Eq. 4** in lines 137-170 in the revised manuscript and in lines 46-54 in supplementary materials, which are highlighted in blue color. All the issues raised by the reviewer have been addressed. Thank the reviewer for the valuable feedback.

Comment 8: Same problems on the descriptions of equations on the supplementary materials. In Eq. (S2), how does a 2x1 vector equals a 3x1 vector? Incident polarization has no subscript, but Eq. (s2) has.

Reply: We thank the reviewer for pointing out this issue. In the derivation of **Eq. s2** (**Eq. s11** in the revised supplementary materials), the polarization expression follows a universal format. $E_{out}(\alpha_{jk}, \beta_{jk})$ represents a zero-dimensional output quantity.

Although the derivation is carried out in this manner, it's important to note that the final simplified forms are equivalent and are solely used to analyze the degrees of freedom.

To provide a clearer understanding, the derivation process is as follows:

$$\begin{aligned}
E_{out}(\alpha_{jk}, \beta_{jk}) &= \begin{bmatrix} \cos \alpha_k \\ \sin \alpha_k \cdot e^{i\beta_k} \end{bmatrix}^\dagger \cdot \begin{bmatrix} a_A e^{i\varphi_A} & a_B e^{i\varphi_B} \\ a_B e^{i\varphi_B} & a_A e^{i\varphi_D} \end{bmatrix} \cdot \begin{bmatrix} \cos \alpha_j \\ \sin \alpha_j \cdot e^{i\beta_j} \end{bmatrix} \\
&= \begin{bmatrix} \cos \alpha_k & \sin \alpha_k \cdot e^{i\beta_k} \end{bmatrix} \cdot \begin{bmatrix} a_A e^{i\varphi_A} \cdot \cos \alpha_j + a_B e^{i\varphi_B} \cdot \sin \alpha_j \cdot e^{i\beta_j} \\ a_B e^{i\varphi_B} \cdot \cos \alpha_j + a_A e^{i\varphi_D} \cdot \sin \alpha_j \cdot e^{i\beta_j} \end{bmatrix} \\
&= \cos \alpha_j \cdot \cos \alpha_k \cdot a_A e^{i\varphi_A} + (\cos \alpha_j \cdot \sin \alpha_k \cdot e^{-i\beta_k} + \sin \alpha_j \cdot \cos \alpha_k \cdot e^{i\beta_j}) \cdot a_B e^{i\varphi_B} \\
&\quad + \sin \alpha_j \cdot \sin \alpha_k \cdot e^{i(\beta_j - \beta_k)} \cdot a_A e^{i\varphi_D} \\
&= \begin{bmatrix} \cos \alpha_j \cdot \cos \alpha_k & \cos \alpha_j \cdot \sin \alpha_k \cdot e^{-i\beta_k} + \sin \alpha_j \cdot \cos \alpha_k \cdot e^{i\beta_j} & \sin \alpha_j \cdot \sin \alpha_k \cdot e^{i(\beta_j - \beta_k)} \end{bmatrix} \cdot \begin{bmatrix} a_A e^{i\varphi_A} \\ a_B e^{i\varphi_B} \\ a_D e^{i\varphi_D} \end{bmatrix}
\end{aligned}$$

We have ensured that the descriptions of the equations in the supplementary materials are consistent and clear. The updated derivation process and the clear definition of the involved physical quantities should address the issues raised by the reviewer.

Comment 9: How can we arrive Eq. (S3) from Eq. (s2)?? It claims three non-

orthogonal input polarizations. What are the expressions of those three polarizations? are they subscripted by number 1, 2, 3? But the subscript 2 was used by the analyzer polarization. So it is very messy and unclear. The whole deducing self-consistent procedure should be provided to reach the publication level.

Reply: We thank the reviewer for this important comment. To address the issue raised, we have provided a detailed derivation of **Eq. s3** from **Eq. s2** in the revised supplementary materials (**Eq. s12** from **Eq. s11** in the revised supplementary materials).

In **Eq. s11**, our goal is to derive a universal form for the polarization channels. By substituting the subscripts j and k with the values corresponding to the three cyclic nonorthogonal polarizations (i.e., $j=1$ and $k=2$, $j=2$ and $k=3$, $j=3$ and $k=1$), we obtain three distinct formulas for **Eq. s11**. Each formula represents a channel controlled by an input $\mathbf{p}_j = \begin{bmatrix} \cos\alpha_j \\ \sin\alpha_j \cdot e^{i\beta_j} \end{bmatrix}$ and output $\mathbf{p}_k = \begin{bmatrix} \cos\alpha_k \\ \sin\alpha_k \cdot e^{i\beta_k} \end{bmatrix}$. By arranging these formulas in the form of a column vector, we derive **Eq. s12**.

To provide a clear understanding, the derivation process is shown below:

For three cyclic nonorthogonal input and output polarizations ($j=1, 2, 3$ and $k=2, 3, 1$), **Eq. s11** becomes:

$$O_1 = \begin{bmatrix} \cos\alpha_1 \cdot \cos\alpha_2 & \cos\alpha_1 \cdot \sin\alpha_2 \cdot e^{-i\beta_2} + \sin\alpha_1 \cdot \cos\alpha_2 \cdot e^{i\beta_1} & \sin\alpha_1 \cdot \sin\alpha_2 \cdot e^{i(\beta_1-\beta_2)} \end{bmatrix} \cdot \begin{bmatrix} a_A e^{i\varphi_A} \\ a_B e^{i\varphi_B} \\ a_D e^{i\varphi_D} \end{bmatrix}$$

$$O_2 = \begin{bmatrix} \cos\alpha_2 \cdot \cos\alpha_3 & \cos\alpha_2 \cdot \sin\alpha_3 \cdot e^{-i\beta_3} + \sin\alpha_2 \cdot \cos\alpha_3 \cdot e^{i\beta_2} & \sin\alpha_2 \cdot \sin\alpha_3 \cdot e^{i(\beta_2-\beta_3)} \end{bmatrix} \cdot \begin{bmatrix} a_A e^{i\varphi_A} \\ a_B e^{i\varphi_B} \\ a_D e^{i\varphi_D} \end{bmatrix}$$

$$O_3 = \begin{bmatrix} \cos\alpha_3 \cdot \cos\alpha_1 & \cos\alpha_3 \cdot \sin\alpha_1 \cdot e^{-i\beta_1} + \sin\alpha_3 \cdot \cos\alpha_1 \cdot e^{i\beta_3} & \sin\alpha_3 \cdot \sin\alpha_1 \cdot e^{i(\beta_3-\beta_1)} \end{bmatrix} \cdot \begin{bmatrix} a_A e^{i\varphi_A} \\ a_B e^{i\varphi_B} \\ a_D e^{i\varphi_D} \end{bmatrix}$$

When O_1, O_2, O_3 are combined into a column vector, we obtain:

$$\begin{bmatrix} O_1 \\ O_2 \\ O_3 \end{bmatrix} = \begin{bmatrix} \cos\alpha_1 \cdot \cos\alpha_2 & \cos\alpha_1 \cdot \sin\alpha_2 \cdot e^{-i\beta_2} + \sin\alpha_1 \cdot \cos\alpha_2 \cdot e^{i\beta_1} & \sin\alpha_1 \cdot \sin\alpha_2 \cdot e^{i(\beta_1-\beta_2)} \\ \cos\alpha_2 \cdot \cos\alpha_3 & \cos\alpha_2 \cdot \sin\alpha_3 \cdot e^{-i\beta_3} + \sin\alpha_2 \cdot \cos\alpha_3 \cdot e^{i\beta_2} & \sin\alpha_2 \cdot \sin\alpha_3 \cdot e^{i(\beta_2-\beta_3)} \\ \cos\alpha_3 \cdot \cos\alpha_1 & \cos\alpha_3 \cdot \sin\alpha_1 \cdot e^{-i\beta_1} + \sin\alpha_3 \cdot \cos\alpha_1 \cdot e^{i\beta_3} & \sin\alpha_3 \cdot \sin\alpha_1 \cdot e^{i(\beta_3-\beta_1)} \end{bmatrix} \cdot \begin{bmatrix} a_A e^{i\varphi_A} \\ a_B e^{i\varphi_B} \\ a_D e^{i\varphi_D} \end{bmatrix} \quad (\text{s12})$$

In this way, we derive **Eq. s12**. To reflect this process in the revised supplementary materials, we have added the detailed description and highlighted it in blue in lines 148-149.

Comment 10: “With a polarization angle α selected at 1 rad interval, and considering the three variables in the equations, the matrix's rank needs to be calculated 5,832,000 times. After accounting for the distinct polarizations, 5,783,160 cases remain.” How to arrive at those numbers: 5,832,000, 5,783,160? It is never mentioned, which makes the manuscript very incomplete.

Reply: We thank the reviewer for this comment. We understand the importance of clearly explaining how the numbers 5,832,000 and 5,783,160 were derived. Due to the difficulty in precisely representing the rank of $O = KX$ obtained from **Eq. s12** in the revised supplementary material, we opted to demonstrate it through numerical

validation.

Taking linear polarization as an example, with each polarization angle separated by 1 degree, there are 180 possible values for each polarization angle (from 0 to 179 degrees). Therefore, the total number of combinations for three polarizations is $180 \times 180 \times 180 = 5,832,000$.

In cases where any two of the three polarizations must be different, the number of valid combinations is calculated as $180 \times 179 \times 178 = 5,735,160$. This accounts for the distinct polarizations required for the analysis. We acknowledge a typing error in the original manuscript where 5,735,160 was incorrectly written as 5,783,160. This has been corrected.

We have updated **Supplementary Note 4** to include these calculations and detailed explanations, which are highlighted in blue in lines 156-160 for clarity. Thank the Reviewer for the feedback, which helps us improve the completeness and accuracy of our manuscript.

Comment 11: The last paragraph is a repeated paragraph on Supplementary Note 1.

Reply: Thank the reviewer for the careful reading effort. We have deleted the repeated paragraph in the revised supplementary materials.

Comment 12: In their present result of the multiple channels of multiplexing, it seems that the choice of the input-output polarization states obey a cyclic rule. But why should we chose this way is not clearly stated in the design strategy. Are other way of polarization pair selection invalid for this approach?

Reply: We thank the reviewer for this insightful comment. In principle, for multi-channel multiplexing, whether it involves linear, elliptical, or circular polarization, there are no specific restrictions on the choice of input and output polarizations, apart from the requirement that neighboring polarizations must be distinct.

In our experiment, to fully represent the coverage of the polarization states, we chose a cyclic rule for the input-output polarization pairs, specifically using end-to-end circular polarization representation. This choice was made to ensure comprehensive coverage and effective demonstration of the polarization states.

To avoid any misunderstanding, we have added the following statement in the revised supplementary materials: 'In order to represent the full coverage of the polarization states in the main text, we choose the input and output polarization which follows a circular rule. Actually, there is no specific requirement for the choice of operation polarization states.' This addition can be found in lines 162-165 on page 9 of the revised supplementary materials.

Comment 13: It concludes that the maximum degree of freedom is represented by the upper or lower nonzero triangle matrix, but why is that is not clearly stated.

Reply: We thank the reviewer for this insightful comment. In this context, 'degree of freedom' refers to the number of available channels. In the derivation from **Eq. 3** to **Eq. 4**, the input and output polarizations can be arbitrarily selected. However, due to the reciprocity of the metasurface, the channels O_{ij} and O_{ji} are the same.

To illustrate, consider two cases: channel $O_{12}(x, y)$, which corresponds to input polarization p_1 and output polarization p_2 , and channel $O_{21}(x, y)$, which corresponds to input polarization p_2 and output polarization p_1 . Due to the reciprocity of the metasurface, as discussed in **Supplementary Note 1**, $O_{12}(x, y) = O_{21}(x, y)$. Therefore, these two cases count as one channel.

Essentially, this means that the number of channels that can be reused depends on the dimensions of the upper or lower triangular nonzero matrices, which correspond to the user-defined number of input (output) polarization states, denoted as n . In this situation, the multiplexing number is not restricted in theory.

To clearly describe and express this conclusion, we have rewritten the content in the supplementary materials. The previous statement 'Thus, the maximum degree of freedom is represented by the upper (or lower) nonzero triangular matrix, as shown in the main text **Eq. 4**.' is revised to: 'Due to the reciprocity of the metasurface, the polarization multiplexing channels O_{jk} and O_{kj} are the same. Therefore, the maximum number of channels is represented by the upper (or lower) nonzero triangular matrix, as shown in the main text **Eq. 4**. In this formula, n ($n=1,2,3,4,5,6\dots$) refers to the number of pairs of input and output polarization states.' This revised content is highlighted in blue in lines 46-50 on page 3 of the revised supplementary materials.

Comment 14: The implementation of the multiple holograms on the metasurfaces are optimized by vectorial diffraction neural network design. Is it just based on deep learning method on electric computer, not the physical optical neural network that perform the computing all-optically? It is also very confusing.

Reply: We thank the reviewer for this insightful comment. We confirm that the optimization process of the metasurface is implemented using a deep learning method based on an electronic computer. The neural network built in the computer consists of several layers. Once we achieve the optimized values, these are implemented with a single metasurface layer.

Unlike the all-optical deep learning framework known as diffractive deep neural networks, as reported in the reference DOI: 10.1126/science.aat8084, our approach integrates the polarization dimension of light into a single layer of the metasurface, rather than using a series of cascading optical elements. While our metasurface design approach is consistent with the reported implementation of all-optical deep learning frameworks using electronic neural networks, it is important to clarify that our optimization process is conducted electronically.

To avoid any confusion, we have added the following content to the main text: 'It should be noted that the definition of the diffraction neural network follows that outlined in Ref. 27. However, our approach differs in that we integrate the polarization dimension of light into a single-layer metasurface, rather than using a series of cascading optical elements. The optimization is conducted using electronic neural networks.' This addition is highlighted in blue in lines 193-197 on page 8 of the revised manuscript.

Comment 15: In the optimization model for the holographic image of Fig. 2a, not

sufficient information about the network including its input and output, and the network layer number as well as node number, and how the optimization of multiple holographic images are assigned to the input and output polarizations. And the connection between the optimization network and their developed high-dimensional Jones Matrix of Eq. (4) is not given, which make it hard to evaluate its validity and significance of the proposed approach.

Reply: We thank the reviewer for this important comment. In **Fig. 2a**, the input is a plane wave with a certain polarization, and the output is the modulated light in another polarization. The process involves calculating the loss with the target pattern, optimizing parameters using the Adam optimizer, and proceeding to the next round of the optimization loop.

Supplementary Note 2 provides a detailed description of the network implementation process. The metaatoms, serving as the fundamental nodes, undergo continuous optimization within the neural network. Connections between individual nodes are established through connection weights, which serve as optimization parameters, linking them to the overall network. These connection weights determine the influence of each node, i.e., the metaatom, on the overall network. By adjusting the optimization parameters of each node, comprehensive optimization of the entire network can be achieved.

To streamline the experiment, we consolidate the hidden layers in the network into a single layer to facilitate the design and implementation of the single-layer metasurface. Once the design is finalized, the fabricated single-layer metasurface will operate in an all-optical manner based on the configured settings.

According to the reviewer's suggestion, we have added the following content to **Supplementary Note 2** to provide more details about the nodes: 'In the training of three-channel linear polarization multiplexing, we set the transmittance of the metaatoms to 1, meaning that the training parameters only include phase and rotation angle. Given that the designed single-layer metasurface consists of 400×400 metaatoms, the total number of node parameters is $3 \times 400 \times 400 = 4.8 \times 10^5$. For the training of three-channel circular and elliptical polarization multiplexing, we include the transmittance of the metaatoms, resulting in a total of $5 \times 400 \times 400 = 8 \times 10^5$ node parameters. In the training of 55-channel multiplexing, there are a total of $5 \times 800 \times 800 = 3.2 \times 10^6$ node parameters involved. Similar calculations apply to other cases in the main text.' This content is highlighted in blue in lines 112-120 on page 6 of **Supplementary Note 2**.

Additionally, the connection between the optimization network and the high-dimensional Jones Matrix is clarified in lines 71-78 in the revised supplementary materials. The optimization process in the neural network directly influences the Jones matrix elements, which represent the polarization states and their transformations.

Thank you for your valuable feedback, which has helped us clarify and improve the comprehensiveness of our manuscript.

Comment 16: In addition, only 3 channel multiplexing are detailed discussed in Figure 2, Figure 3, Figure 4, Figure S1, S2, S3 and so on. The data for the 55 channel

multiplexing is too little, only captured image patterns on Figure 5, without its loss function evolution, simulation results, the crosstalk analysis and so on. Those data is very important to evaluate the quality of the proposed approach.

Reply: We thank the reviewer for this valuable comment. In response to your suggestion, we have added **Supplementary Note 9** to provide a detailed discussion of the 55-channel multiplexing, including simulation results, crosstalk analysis, and loss function evolution. These additional data are crucial for evaluating the quality of the proposed approach.

Please refer to the revised **Supplementary Note 9** for the detailed contents and figure. These additions ensure a comprehensive evaluation of the 55-channel multiplexing and enhance the overall robustness and clarity of our work.

Comment 17: And very importantly, as the main work take advantage of the collection effect rather than individual meta-atom. The dimension of their developed higher dimension Jones matrix in Eq. (4) should be ambiguously present. At its present form, $O(x, y)$ is of the dimension of the number of polarization numbers n , it is still associated with a single meta-atom with different polarization responses. As it is spatially varying with coordinates x, y , so should the dimension of the matrix be $n_{\text{meta}}*n$, where n_{meta} is the number of the meta-atoms in one direction. In this formalism, what is the actual form of the matrix? And what is the relation between Eq. (4) and Eq. (3), is Eq. (4) derived from Eq. (3), one can hardly find a clue.

Reply: We thank the reviewer for this important comment. As mentioned in our responses to previous comments, we have thoroughly revised the derivation process in both the manuscript and supplementary materials to improve readability and coherence.

We want to emphasize that we utilize the spatially varied eigen-polarization states of the metaatoms to successfully reconstruct globally nonorthogonal channels that exhibit minimal crosstalk. This is an overall effect of the metasurface and is independent of the number of metaatoms n_{meta} in a particular direction.

To avoid misunderstandings, we have redesigned the form of the formulas and rewritten the derivation process. In **Eq. 4** of the revised manuscript, the number of channels is related to the Jones matrix dimension, which is determined by the designed polarization number n , and not dependent on the number of metaatoms.

To clearly illustrate the difference between the Jones matrix dimension (which corresponds to the desired multiplexing channel number) and the number of metaatoms, we have provided the specific expression of each element $O_{ij}(x, y)$ in **Eq. 2**. As the input and output polarizations vary, we can obtain multiple sets of $O_{ij}(x, y)$. When these sets are arranged into a matrix, we obtain the response matrix $O(x, y)$ in **Eq. 4**. The global response of nonorthogonal polarization channel is then the integral of **Eq. 4** with respect to position, $\Omega = \iint O(x, y) dx dy$. Besides, the content 'It's worth noting that we utilize the spatially varied eigen-polarization states of the metaatoms to successfully reconstruct globally nonorthogonal channels that exhibit minimal crosstalk.' is added in lines 164-170 on page 7 in the revised manuscript to avoid misleading. We believe these modifications provide a clear understanding of the relationship between **Eq. 3** and **Eq. 4** and highlight the role of the metasurface's collective effect.

Thank you for your valuable feedback, which has helped us clarify and enhance the comprehensiveness of our manuscript.

Response to Reviewer #3

Overall comment: The manuscript by Wang et al. presents a new approach to encoding multiple holographic images in metasurfaces by capitalizing nonorthogonal polarization multiplexing. By controlling the input/output polarizations, the authors demonstrated an unprecedented number of 55 holographic images with one metasurface. Overall, I find the results reported in this paper interesting and I think it deserves to be published in Nature Communications. But before I recommend its acceptance, several points need to be addressed by implementing a revision.

Reply: We thank the reviewer for the positive evaluation of our work and for acknowledging the significance of our approach in encoding multiple holographic images using nonorthogonal polarization multiplexing. We appreciate your recognition of the novelty and potential impact of demonstrating 55 holographic images with a single metasurface.

We have carefully considered all the points raised and have implemented revisions to address each comment. Detailed responses to each of your specific comments are provided below. We believe that these revisions have enhanced the clarity, comprehensiveness, and robustness of our manuscript.

Thank you for your valuable feedback, which has helped us improve the quality of our work. We hope that the revised manuscript meets the high standards of Nature Communications and look forward to your favorable consideration for publication.

Comment 1: The authors reformed the response of the metasurface to an $n \times n$ matrix to facilitate analysis under multiple input-output polarization channels. This form of analysis is good, but I would like to discuss with the authors whether the nature of the metasurfaces' light manipulation capability changes. It seems that the components in the extended matrix are related to each other, which is different from the completely decoupled elements in the 2×2 Jones matrix. Optimization for multiple holographic images can be done, but I wonder if it is possible to implement other types of functionalities, e.g., a continuous zoom lens by varying the input-output polarization. Please comment on this.

Reply: We thank the reviewer for this open-minded question. Our functionality is achieved by controlling individual metaatoms, maintaining the overall effect without altering the fundamental mechanism of light manipulation. In simpler terms, traditional control methods viewed manipulation from the perspective of individual metaatom or local regions, where each part operated relatively independently. Our work represents a deeper exploration by considering the perspective of the whole metasurface, particularly through the coherent control of multiple metaatoms' eigen-polarizations.

The extended $n \times n$ matrix we introduced does indeed establish connections between the components, which differs from the completely decoupled elements in a traditional 2×2 Jones matrix. This approach enables us to achieve coordinated overall effects while ensuring synchronization among different parts of the metasurface.

As for the potential to implement other types of functionalities, such as a continuous zoom lens by varying the input-output polarization, our method theoretically allows for

such implementations. By precisely controlling the eigen-polarizations of the metaatoms and optimizing the metasurface's global response, we could potentially develop functionalities beyond holography, including adaptive lenses and other dynamic optical devices.

To reflect this in the revised manuscript, the content 'As for the potential to implement other types of functionalities, such as a continuous zoom lens by varying the input-output polarization, our method theoretically allows for such implementations. By precisely controlling the eigen-polarizations of the metaatoms and optimizing the metasurface's global response, we could potentially develop functionalities beyond holography, including adaptive lenses and other dynamic optical devices.' is added in lines 377-382 with blue color to discuss its versatile functionalities.

Thank you for your insightful comment, which has allowed us to further elaborate on the capabilities and potential applications of our approach.

Comment 2: The holographic efficiency is an important characterization value for the performance of meta-holograms, but it seems that the authors do not mention it in the manuscript. Please add these data in the main manuscript or supplementary information.

Reply: We thank the reviewer for this important comment. As the reviewer suggests, we have calculated the holographic efficiency based on the experimental results and added the related content and figures in the revised supplementary materials.

The method for calculating the holographic efficiency is as follows:

where E_{holo} refers to the measured total energy of the hologram pattern, and E_{total} refers to the total incident energy received by the metasurface area.

For the tri-fold cyclic nonorthogonal linear polarization asymmetric holography shown in **Fig. 3**, the measured holographic efficiencies are 26.04% ($0^\circ \rightarrow 60^\circ$ polarized channel), 25.25% ($60^\circ \rightarrow 120^\circ$ polarized channel), and 27.83% ($120^\circ \rightarrow 180^\circ$ polarized channel) respectively.

The measured efficiencies of the Metasurface 1 in **Fig. 4** are 19.13%, 22.16%, and 20.12%, corresponding to the letters 'A', 'B', and 'C', respectively, and the Metasurface 2 are 22.08%, 24.61%, and 21.4% in that order.

The measured efficiencies of the 55 nonorthogonal holographic channels are shown in **Supplementary Fig. 10**. Relevant results can be found in the revised manuscript and the revised supplementary materials.

We appreciate the reviewer's suggestion, which has helped us provide a more comprehensive characterization of the performance of our meta-holograms.

Fig. r9. The measured efficiencies of 55 nonorthogonal holographic channels.

Comment 3: The authors pointed out the lower noise level in the real experimental images compared to those in the simulation. Could the author provide a rationale for this phenomenon?

Reply: We thank the reviewer for this comment. We apologize for the typo that led to a misunderstanding. The intended message was to highlight the comparability of the observed results in the experiments to those in the simulations.

To correct this, we have modified the content in the revised manuscript to: '**It can be seen that the measured results agree well with the simulations.**' This correction can be found in lines 255-257, highlighted in blue.

Thank you for pointing out this error, which has allowed us to improve the clarity and accuracy of our manuscript.

Comment 4: The authors mentioned the use of focal plane mismatch for the design in Fig. 5. Does this mean that the images in Fig. 5c are not captured at the same spatial position?

Reply: We thank the reviewer for this comment. The focal plane mismatch in the design of Fig. 5 refers to the adjustment of the focal plane at different z-axis positions to achieve optimal clarity and effectiveness in image capture. This means that the images in Fig. 5c are indeed captured at slightly different spatial positions along the z-axis to ensure the best possible resolution and image quality for each holographic pattern.

To reflect this in the revised manuscript, the content '**It's worth noting that the images in Fig. 5c are captured at slightly different spatial positions along the z-axis to ensure the best possible resolution and image quality for each holographic pattern.**' is added in lines 340-342 with blue color.

Thank you for your insightful question, which has helped us clarify this aspect of our experimental design.

Comment 5: In the current experiment, the authors showcased the maximum number of 55 independent holographic images. I would like to know the factors that impede encoding more images through the proposed approach in this paper. What is the theoretical limit and is it necessary to find a balance between different holographic performance metrics in real experiments? I would suggest the authors provide

perspectives on this issue and incorporate the discussion in the manuscript for the convenience of readers.

Reply: We thank the reviewer for this thought-provoking comment. As we have proved in **Supplementary Notes 1 and 4**, the nonzero analytical solution of the derived formulas indicates that the largest rank of the coefficient matrix of the Jones matrix is 3. This means that the maximum number of channels for nonorthogonal polarization multiplexing without compromising other dimensions is three. Beyond three channels, crosstalk becomes significant, and different methods must be employed to extend the multiplexing channels.

Our approach involves the use of a controllable local eigen-polarization modulation mechanism, as well as optimization through a vectorial diffraction neural network. Additionally, methods such as noise assistance, as described in Ref. 17, can also be utilized to further extend the number of multiplexing channels.

To achieve more multiplexing channels, the metaatoms must possess stronger control abilities, such as enhanced phase coverage, more design degrees of freedom, and other properties. This necessitates metaatoms with larger depth-to-width ratios, higher refractive index differences, more complex shapes, or additional metasurface layers to expand their controlling capabilities. These factors can influence the efficiency, crosstalk, and the number of achievable multiplexing channels.

To reflect this discussion in the revised manuscript, we have added the following content: 'The nonzero analytical solution of the derived formulas in **Supplementary Notes 1 and 4** indicates that the largest rank of the coefficient matrix of the Jones matrix is 3. This means that for given metaatoms with similar profiles, the maximum number of channels for nonorthogonal polarization multiplexing without compromising other dimensions is three. Beyond three channels, crosstalk becomes significant, and different methods must be employed to extend the multiplexing channels. Our approach involves the use of a controllable local eigen-polarization modulation mechanism, as well as optimization through a vectorial diffraction neural network. Additionally, methods such as noise assistance, as described in Ref. 17, can also be utilized to further extend the number of multiplexing channels. To achieve more multiplexing channels, it imposes a higher requirement on the metaatoms to possess stronger control abilities, such as enhanced phase coverage, more design degrees of freedom, etc. This necessitates metaatoms with larger depth-to-width ratios, higher refractive index differences, more complex shapes, or additional metasurface layers to expand their controlling capabilities. These factors can influence the efficiency, crosstalk, and the number of achievable multiplexing channels.' This addition is highlighted in blue in lines 361-376 of the revised manuscript.

Thank the Reviewer for this valuable suggestion, which has helped us improve the comprehensiveness and clarity of our manuscript.

Comment 6: Holography and polarization multiplexing are important topics of concern to the metasurface community. The following references can be added to provide a more thorough background to the readers.

<https://www.nature.com/articles/s41467-019-12637-0>

<https://link.springer.com/article/10.1186/s43593-023-00054-2>

<https://www.nature.com/articles/s41467-022-35313-2>

<https://link.springer.com/article/10.1186/s43593-024-00061-x>

<https://www.nature.com/articles/s41563-023-01531-2>

<https://link.springer.com/article/10.1186/s43593-021-00002-y>

Reply: We thank the reviewer for this valuable comment and for suggesting additional references to enhance the background of our manuscript. We agree that providing a more thorough background on holography and polarization multiplexing is beneficial for readers.

We have reviewed the suggested references and incorporated them into the revised manuscript to provide a more comprehensive background on the topics of concern to the metasurface community. Together with four additional papers, the above references have been added and discussed in the introduction part in the revised manuscript to provide readers with a more thorough background on the current advancements and applications of holography and polarization multiplexing in metasurfaces.

Comment 7: I would like to provide some suggestions for several minor issues to further improve the quality of the manuscript:

- I got confused when I saw "vectorial optical diffraction neural network" in the text. In conventions, we speak about optical neural networks when some tasks are done physically by light diffraction, which seems not the case in this work."vectorial diffraction neural network" also appears in the text. Please unify the expressions.

- Please provide the explicit definition of all parameters in the equations, e.g., O_{ij} in Eq.2 and a_{ij} and Eq. 3.

- Please add more description about how Fig. 1b is related to the formulas in the "Design principle" section.

- Please provide the full names of the abbreviations (LP, PR, FPA) in Fig. 2e.

- Please add the labels in the bottom axes so that data in Fig. 3d would have clear meanings.

- Please check the description of the labels in Figure 3. It should be (e)-(g) rather than (e)-(f) in the caption.

Reply: We thank the reviewer for the detailed and constructive suggestions to improve the quality of our manuscript. We have seriously considered your comments and revised the manuscript accordingly.

-We appreciate the suggestion regarding the terminology. We have standardized the expression by revising 'vectorial optical diffraction neural network' to 'vectorial diffraction neural network' throughout the manuscript.

-We have carefully reviewed the manuscript and added explicit definitions for all parameters in the equations, including O_{ij} in Eq. 2 and a_{ij} in Eq. 3.

-To clarify the connection between Fig. 1b and the formulas in the 'Design principle' section, we have added additional formulas and provided more descriptions.

-We have added the full names of the abbreviations in Fig. 2e. The content now reads: 'LP stands for linear polarization, PR for phase retarder, and FPA for focal plane array.' This addition is highlighted in blue in lines 182-183 of the revised

manuscript.

-The labels in the bottom axes of **Fig. 3d** have been added and description is also added to the illustrations to provide clear meanings for the data.

Fig. r6. The crosstalk of the tri-fold cyclic nonorthogonal linear polarization multiplexing in **Fig. 3**.

-We have corrected the description of the labels in **Fig. 3**. It should indeed be (e)-(g) rather than (e)-(f). The content has been updated to: 'e-g Longitudinal intensity statistics...' in line 234 of the revised main text.

Thank the Reviewer again for the valuable feedback, which has helped us improve the clarity and quality of our manuscript.

REVIEWERS' COMMENTS

Reviewer #1 (Remarks to the Author):

The authors have addressed my concerns carefully. Although their approach cannot break the limitation of 3 uncoupled information channels, as reported by many other references, they do show more-than-three holographic channels can be established by polarization multiplexing and space multiplexing, assisted by NN training.

Reviewer #2 (Remarks to the Author):

The authors have made substantial and thorough revisions to their manuscript, particularly in the sections that explain the underlying principles and the derivation of equations. As a result, the physical concepts have been elucidated with much greater clarity, and the step-by-step derivation of the formulas is now presented in a very comprehensible manner. Additionally, the paper has achieved the highest reported polarization multiplexing channels, highlighting the work's significance and innovation. The improvements have significantly enhanced the readability and scientific rigor of the paper.

The revised manuscript effectively addresses the concerns raised in the previous review, and no further issues remain. I am confident that this paper represents a valuable contribution to the field and meets the high standards required for publication in Nature Communications. Therefore, I am pleased to recommend that this paper be accepted for publication.

Reviewer #3 (Remarks to the Author):

I read through the reply from the authors to my comments as well as other referees. I found my concerns have been well addressed, and also quite reasonable responses and changes made toward other referees' questions. Hence I would like to recommend it for publication.

Response to Reviewer #1

Overall Comment: The authors have addressed my concerns carefully. Although their approach cannot break the limitation of 3 uncoupled information channels, as reported by many other references, they do show more-than-three holographic channels can be established by polarization multiplexing and space multiplexing, assisted by NN training.

Reply: Thank the reviewers for your detailed review and feedback on our paper. We are pleased that the reviewers acknowledged our responses and improvements to the questions you raised.

We have further revised the manuscript to address the editorial requests. We hope this study can provide valuable reference and enlightenment for future research in this field.

Response to Reviewer #2

Overall Comment: The authors have made substantial and thorough revisions to their manuscript, particularly in the sections that explain the underlying principles and the derivation of equations. As a result, the physical concepts have been elucidated with much greater clarity, and the step-by-step derivation of the formulas is now presented in a very comprehensible manner. Additionally, the paper has achieved the highest reported polarization multiplexing channels, highlighting the work's significance and innovation. The improvements have significantly enhanced the readability and scientific rigor of the paper.

The revised manuscript effectively addresses the concerns raised in the previous review, and no further issues remain. I am confident that this paper represents a valuable contribution to the field and meets the high standards required for publication in Nature Communications.

Therefore, I am pleased to recommend that this paper be accepted for publication.

Reply: Thank you for your valuable time and effort. We are grateful to the reviewer for your detailed review and constructive comments on our manuscript. It is these comments that help us to further improve the paper.

We have further revised the manuscript to address the editorial requests. We hope this study can provide valuable reference and enlightenment for future research in this field.

Response to Reviewer #3

Overall Comment: I read through the reply from the authors to my comments as well as other referees. i found my concerns have been well addressed, and also quite reasonable responses and changes made toward other referees' questions. Hence i would like to recommend it for publication.

Reply: We sincerely thank the reviewer for your detailed review and positive feedback on our paper. We are pleased to know that we have made a reasonable response and the corresponding changes, and have fully addressed your concerns.

We have further revised the manuscript to address the editorial requests. We hope this study can provide valuable reference and enlightenment for future research in this field.